# Self-Play Preference Optimization for Language Model Alignment

**Yue Wu**[1*]**, Zhiqing Sun**[2*]**, Huizhuo Yuan**[1*]**, Kaixuan Ji**[1]**, Yiming Yang**[2]**, Quanquan Gu**[1†]

[1]Department of Computer Science, University of California, Los Angeles

[2]Language Technologies Institute, Carnegie Mellon University

`{ywu,hzyuan,qgu}@cs.ucla.edu, {zhiqings,yiming}@cs.cmu.edu`

## Abstract

Standard reinforcement learning from human feedback (RLHF) approaches relying on parametric models like the Bradley-Terry model fall short in capturing the intransitivity and irrationality in human preferences. Recent advancements suggest that directly working with preference probabilities can yield a more accurate reflection of human preferences, enabling more flexible and accurate language model alignment. In this paper, we propose a self-play-based method for language model alignment, which treats the problem as a constant-sum two-player game aimed at identifying the Nash equilibrium policy. Our approach, dubbed *Self-Play Preference Optimization* (SPPO), utilizes iterative policy updates to provably approximate the Nash equilibrium. Additionally, we propose a new SPPO objective which is both strongly motivated by theory and is simple and effective in practice. In our experiments, using only 60k prompts (without responses) from the UltraFeedback dataset and without any prompt augmentation, by leveraging a pre-trained preference model PairRM with only 0.4B parameters, SPPO can obtain a model from fine-tuning Mistral-7B-Instruct-v0.2 that achieves the state-of-the-art length-controlled win-rate of 28.53% against GPT-4-Turbo on AlpacaEval 2.0. It also outperforms the (iterative) DPO and IPO on MT-Bench, Arena-Hard, and the Open LLM Leaderboard. Starting from a stronger base model Llama-3-8B-Instruct, we are able to achieve a length-controlled win rate of 38.77%. Notably, the strong performance of SPPO is achieved without additional external supervision (e.g., responses, preferences, etc.) from GPT-4 or other stronger language models.

## 1 Introduction

Large Language Models (LLMs) (e.g., Ouyang et al., 2022; OpenAI et al., 2023), have shown remarkable capabilities in producing human-like text, fielding questions, and coding. Despite their advancements, these models encounter challenges in tasks requiring high levels of reliability, safety, and ethical alignment. To address these challenges, Reinforcement Learning from Human Feedback (RLHF), also known as Preference-based Reinforcement Learning (PbRL), presents a promising solution. This framework for policy optimization, highlighted in works by Christiano et al. (2017) and recently in Ouyang et al. (2022), has led to significant empirical success in fine-tuning instruction-following LLMs, making them more aligned with human preferences and thus more helpful.

Most existing approaches to RLHF rely on either explicit or implicit reward models. Taking InstructGPT (Ouyang et al., 2022) as an example, a reference policy $\pi_{\text{ref}}$ is first established, typically from supervised pre-training or instruction-based (supervised) fine-tuning. An explicit reward function is obtained by training a reward model based on human preference feedback data, employing the Bradley-Terry (BT) model (Bradley & Terry, 1952). Subsequently, reinforcement learning algorithms such as Proximal Policy Optimization (Schulman et al., 2017, PPO) are used to fine-tune the reference LLM $\pi_{\text{ref}}$ by maximizing the reward function. The reward model provides a "reward score" $r(\mathbf{y}; \mathbf{x})$ for the given response $\mathbf{y}$ and prompt $\mathbf{x}$, approximately reflecting how humans value these responses. More recently, methods like Direct Preference Optimization (Rafailov et al., 2024b, DPO) have been introduced. These methods forgo the training of a separate reward model but still

---

[*]Equal Contribution

[†]Corresponding Author

fundamentally adhere to the reward maximization objective and are determined by parametric models such as the BT model.

These models presuppose a monotonous and transitive relationship among preferences for different choices. However, empirical evidence suggests otherwise. For instance, Tversky (1969) observed human decisions can be influenced by different factors and exhibit inconsistency. Such observations indicate that human preferences do not always adhere to a single, value-based hierarchy and can even appear irrational, such as exhibiting loops in preference relations. For LLMs, another motivating evidence is that Munos et al. (2023) has empirically shown that directly predicting the pairwise preference can achieve higher accuracy than predicting the preference via a BT-based reward model.

To address the inconsistency in human preference, researchers have proposed to work directly with the preference probability and design algorithms that can more flexibly represent human preferences (Lou et al., 2022; Wu et al., 2023) in the ranking or bandit setting. Recently, an emerging line of work (Wang et al., 2024; Munos et al., 2023; Swamy et al., 2024) also proposed to study RLHF for LLMs under such general preference $\mathbb{P}(\mathbf{y} \succ \mathbf{y}'|\mathbf{x})$, where $\mathbf{y}$ and $\mathbf{y}'$ are two different responses and $\mathbf{x}$ is prompt. The goal is to identify the Nash equilibrium or von Neumann winner of the two-player constant-sum game

$$(\pi^*, \pi^*) = \arg \max_{\pi} \min_{\pi'} \mathbb{E}_{\mathbf{x} \sim \mathcal{X}} \left[ \mathbb{E}_{\mathbf{y} \sim \pi(\cdot|\mathbf{x}), \mathbf{y}' \sim \pi'(\cdot|\mathbf{x})} \left[ \mathbb{P}(\mathbf{y} \succ \mathbf{y}'|\mathbf{x}) \right] \right],$$

where each player is an LLM that outputs responses and aims to maximize its probability of being preferred over its opponent.

Independent from our work, Swamy et al. (2024) proposed Self-play Preference Optimization (SPO)[1] for the same (unregularized) two-player constant-sum game. They provide a general reduction of preference optimization to no-regret online learning for the multi-step Markov Decision Process. When constrained to the bandit setting for LLMs, their proposed algorithmic framework reduces to the famous Hedge algorithm (Freund & Schapire, 1997), which admits the exponential update rule as described in (3.1). To approximately solve the exponential update, Swamy et al. (2024) then proposed to employ typical policy optimization algorithms such as Proximal Policy Optimization (PPO) (Schulman et al., 2017) or Soft Actor-Critic (SAC) (Haarnoja et al., 2018) to maximize the win rate against the reference policy and evaluated the performance of their self-play algorithms in robotic and game tasks. However, it typically requires more effort to apply PPO or SAC to large-scale fine-tuning of LLM and make them work stably. Therefore, it remains unclear how their self-play framework can be applied to large-scale language model alignment efficiently.

In this paper, motivated by these developments mentioned above, we propose a new self-play algorithm that (1) enjoys provable guarantees to solve the two-player constant-sum game; and (2) can scale up to large-scale efficient fine-tuning of large language models. In detail, we formulate the RLHF problem as a constant-sum two-player game. Our objective is to identify the Nash equilibrium policy, which consistently provides preferred responses over any other policy on average. To identify the Nash equilibrium policy approximately, we adopt the classic online adaptive algorithm with multiplicative weights (Freund & Schapire, 1999) as a high-level framework that solves the two-player game. Further, each step of the high-level framework can be approximated by a *self-play* mechanism, where in each iteration the policy is playing against itself in the previous iteration by fine-tuning it on synthetic data that are generated by the policy and annotated by the preference model.

Our contributions are highlighted as follows:

- Starting from the exponential weight update algorithm which provably converges to the Nash equilibrium of the two-player constant-sum game, we propose the *Self-Play Preference Optimization* (SPPO) algorithm for large language model alignment. The algorithm converges to an approximate Nash equilibrium provably and admits a simple form of loss function for easy optimization.

- Unlike the symmetric pairwise loss such as DPO and Identity Preference Optimization (IPO) (Azar et al., 2023), we propose a new optimization objective that does not rely on pairwise comparisons. The new loss objective (3.4), initially driven by game-theoretical concepts, turns out strongly motivated by the policy gradient theory and implicitly encourages the LLM to learn a token-level optimal value function.

---

[1]The SPO framework does not pertain to the efficient fine-tuning of LLMs. Our Self-Play Preference Optimization (SPPO) focuses on LLM alignment and was developed independently. To distinguish it from the SPO framework, we use the abbreviation SPPO.

- Empirically, SPPO significantly enhances the well-aligned Mistral-7B-Instruct-v0.2 and Llama-3-8B-Instruct model, achieving an increase of over 11% on the length-controlled win rate against GPT-4-Turbo on the AlpacaEval 2.0 (Dubois et al., 2024a) test set. Additionally, SPPO exhibits strong generalist abilities across different tasks, including MT-Bench, the Open LLM Leaderboard, and the more recent, more challenging benchmark, Arena-Hard. Unlike iterative DPO/IPO, which tends to show performance decay on other benchmarks when optimized towards the PairRM score, SPPO's performance gain is consistent. Notably, all the strong performances are achieved without external supervision (e.g., responses, preferences, etc.) from GPT-4 or other stronger language models.

Concurrent to our work, several studies, including Direct Nash Optimization (Rosset et al., 2024) and REBEL (Gao et al., 2024) have also explored using either cross-entropy loss or square loss minimization to approximate the exponential update. Specifically, they used the same trick proposed in DPO (Rafailov et al., 2024b) to cancel out the log-partition factor and directly regress on the win-rate difference. However, it is shown theoretically and empirically by Pal et al. (2024) that the pairwise loss may only drive the *relative* likelihood gap to be large, but may not necessarily drive up the likelihood of the preferred responses. Our method instead has a deeper connection to the policy gradient theory and can effectively match the likelihood of the response to its win rate. We postpone the detailed discussion of related works to Appendix A.

## 2 PRELIMINARIES

We consider the preference learning scenario as follows. Given a text sequence (commonly referred to as prompt) $\mathbf{x} = [x_1, x_2, \dots]$, two text sequences $\mathbf{y} = [y_1, y_2, \dots]$ and $\mathbf{y}'$ are generated as responses to the prompt $\mathbf{x}$. An autoregressive language model $\pi$ given the prompt $\mathbf{x}$ can generate responses $\mathbf{y}$ following the probability decomposition

$$\pi(\mathbf{y}|\mathbf{x}) = \prod_{i=1}^{N} \pi(y_i|\mathbf{x}, \mathbf{y}_{<i}).$$

Given the prompt $\mathbf{x}$ and two responses $\mathbf{y}$ and $\mathbf{y}'$, a preference oracle (either a human annotator or a language model) will provide preference feedback $o(\mathbf{y} \succ \mathbf{y}'|\mathbf{x}) \in \{0, 1\}$ indicating whether $\mathbf{y}$ is preferred over $\mathbf{y}'$. We denote $\mathbb{P}(\mathbf{y} \succ \mathbf{y}'|\mathbf{x}) = \mathbb{E}[o(\mathbf{y} \succ \mathbf{y}'|\mathbf{x})]$ as the probability of $\mathbf{y}$ "winning the duel" over $\mathbf{y}'$. The KL divergence of two probability distributions of density $p$ and $q$ is defined as $\mathrm{KL}(p\|q) = \mathbb{E}_{\mathbf{y} \sim p(\mathbf{y})}\left[\log \frac{p(\mathbf{y})}{q(\mathbf{y})}\right]$.

### 2.1 RLHF WITH REWARD MODELS

Christiano et al. (2017) first learn a reward function $r(\mathbf{y}; \mathbf{x})$ following the Bradley-Terry model (Bradley & Terry, 1952). For a prompt-response-response triplet $(\mathbf{x}, \mathbf{y}, \mathbf{y}')$, the Bradley-Terry model specifies the probability of $\mathbf{y}$ being chosen over $\mathbf{y}$ as

$$\mathbb{P}(\mathbf{y} \succ \mathbf{y}'|\mathbf{x}) = \frac{\exp(r(\mathbf{y}; \mathbf{x}))}{\exp(r(\mathbf{y}; \mathbf{x})) + \exp(r(\mathbf{y}'; \mathbf{x}))} = \sigma\big(r(\mathbf{y}; \mathbf{x}) - r(\mathbf{y}'; \mathbf{x})\big), \qquad (2.1)$$

where $\sigma(x) = e^x/(e^x + 1)$ is the logistic function. The reward function associated with the Bradley-Terry model can be estimated by maximizing the log-likelihood $\log \mathbb{P}(\mathbf{y} \succ \mathbf{y}'|\mathbf{x})$. Suppose the true reward function $r(\mathbf{y}; \mathbf{x}))$ is available, Christiano et al. (2017) proposed to solve the following optimization problem with policy optimization algorithms in RL such as PPO (Schulman et al., 2017):

$$\max_{\boldsymbol{\theta}} \mathbb{E}_{\mathbf{x} \sim \mathcal{X}, \mathbf{y} \sim \pi_{\boldsymbol{\theta}}(\cdot|\mathbf{x})}[r(\mathbf{y}; \mathbf{x})] - \eta^{-1}\mathbb{E}_{\mathbf{x} \sim \mathcal{X}}[\mathrm{KL}(\pi_{\boldsymbol{\theta}}(\cdot|\mathbf{x})\|\pi_{\mathrm{ref}}(\cdot|\mathbf{x}))], \qquad (2.2)$$

where $\mathcal{X}$ is the prompt distribution.

Rafailov et al. (2024b) identified that the optimization problem above has a closed-form solution such that for any $\mathbf{y}$,

$$\pi^*(\mathbf{y}|\mathbf{x}) \propto \pi_{\mathrm{ref}}(\mathbf{y}|\mathbf{x}) \exp(\eta r(\mathbf{y}; \mathbf{x})),$$

which can be further converted to the DPO loss for any triplet $(\mathbf{x}, \mathbf{y}_w, \mathbf{y}_l)$ where the winner $\mathbf{y}_w$ is chosen over the loser $\mathbf{y}_l$:

$$\ell_{\mathrm{DPO}}(\mathbf{x}, \mathbf{y}_w, \mathbf{y}_l; \boldsymbol{\theta}; \pi_{\mathrm{ref}}) := -\log \sigma\left(\eta^{-1}\left[\log\left(\frac{\pi_{\boldsymbol{\theta}}(\mathbf{y}_w|\mathbf{x})}{\pi_{\mathrm{ref}}(\mathbf{y}_w|\mathbf{x})}\right) - \log\left(\frac{\pi_{\boldsymbol{\theta}}(\mathbf{y}_l|\mathbf{x})}{\pi_{\mathrm{ref}}(\mathbf{y}_l|\mathbf{x})}\right)\right]\right).$$

## 2.2 RLHF WITH GENERAL PREFERENCE

Following Wang et al. (2024); Munos et al. (2023), we aim to establish RLHF methods without a reward model, as the human preference can be non-transitive (Tversky, 1969). Under a general preference oracle $\mathbb{P}(\mathbf{y} \succ \mathbf{y}'|\mathbf{x})$, we follow Dudík et al. (2015) and aim to identify the *von Neumann winner*. More specifically, the von Neumann winner $\pi^*$ is the (symmetric) Nash equilibrium of the following two-player constant-sum game:

$$(\pi^*, \pi^*) = \arg\max_{\pi} \min_{\pi'} \mathbb{E}_{\mathbf{x} \sim \mathcal{X}} \Big[ \mathbb{E}_{\mathbf{y} \sim \pi(\cdot|\mathbf{x}), \mathbf{y}' \sim \pi'(\cdot|\mathbf{x})} \big[ \mathbb{P}(\mathbf{y} \succ \mathbf{y}'|\mathbf{x}) \big] \Big]. \tag{2.3}$$

In addition, we define the winning probability of one response $\mathbf{y}$ against a distribution of responses $\pi$ as

$$\mathbb{P}(\mathbf{y} \succ \pi|\mathbf{x}) = \mathbb{E}_{\mathbf{y}' \sim \pi(\cdot|\mathbf{x})}[\mathbb{P}(\mathbf{y} \succ \mathbf{y}'|\mathbf{x})],$$

and the winning probability of one policy $\pi$ against another policy $\pi'$ as

$$\mathbb{P}(\pi \succ \pi'|\mathbf{x}) = \mathbb{E}_{\mathbf{y} \sim \pi(\cdot|\mathbf{x})} \mathbb{E}_{\mathbf{y}' \sim \pi'(\cdot|\mathbf{x})}[\mathbb{P}(\mathbf{y} \succ \mathbf{y}'|\mathbf{x})].$$

Furthermore, we define $\mathbb{P}(\pi \succ \pi') = \mathbb{E}_{\mathbf{x} \sim \mathcal{X}}[\mathbb{P}(\pi \succ \pi'|\mathbf{x})]$, where $\mathbf{x}$ is a prompt drawn from the prompt distribution $\mathcal{X}$. The two-player constant-sum game (2.3) can be simplified as

$$(\pi^*, \pi^*) = \arg\max_{\pi} \min_{\pi'} \mathbb{P}(\pi \succ \pi').$$

## 3 SELF-PLAY PREFERENCE OPTIMIZATION (SPPO)

In this section, we introduce the Self-Play Preference Optimization (SPPO) algorithm, derived from the following theoretical framework.

### 3.1 THEORETICAL FRAMEWORK

There are well-known algorithms to approximately solve the Nash equilibrium in a constant-sum two-player game. In this work, we follow Freund & Schapire (1999) to establish an iterative framework that can asymptotically converge to the optimal policy on average. We start with a theoretical framework that conceptually solves the two-player game as follows:

$$\pi_{t+1}(\mathbf{y}|\mathbf{x}) \propto \pi_t(\mathbf{y}|\mathbf{x}) \exp(\eta \mathbb{P}(\mathbf{y} \succ \pi_t|\mathbf{x})), \text{ for } t = 1, 2, \ldots. \tag{3.1}$$

(3.1) is an iterative framework that relies on the multiplicative weight update in each iteration $t$ and enjoys a clear structure. Initially, we have a base policy $\pi_1$ usually from some supervised fine-tuned model. In each iteration, the updated policy $\pi_{t+1}$ is obtained from the reference policy $\pi_t$ following the multiplicative weight update. More specifically, a response $\mathbf{y}$ should have a higher probability weight if it has a higher average advantage over the current policy $\pi_t$.
Equivalently, (3.1) can be written as

$$\pi_{t+1}(\mathbf{y}|\mathbf{x}) = \frac{\pi_t(\mathbf{y}|\mathbf{x}) \exp\big(\eta \mathbb{P}(\mathbf{y} \succ \pi_t|\mathbf{x})\big)}{Z_{\pi_t}(\mathbf{x})}, \tag{3.2}$$

where $Z_{\pi_t}(\mathbf{x}) = \sum_{\mathbf{y}} \pi_t(\mathbf{y}|\mathbf{x}) \exp\big(\eta \mathbb{P}(\mathbf{y} \succ \pi_t|\mathbf{x})\big)$ is the normalizing factor (a.k.a., the partition function). For any fixed $\mathbf{x}$ and $\mathbf{y}$, the ideal update policy $\pi_{t+1}$ should satisfy the following equation:

$$\log\left(\frac{\pi_{t+1}(\mathbf{y}|\mathbf{x})}{\pi_t(\mathbf{y}|\mathbf{x})}\right) = \eta \cdot \mathbb{P}(\mathbf{y} \succ \pi_t|\mathbf{x}) - \log Z_{\pi_t}(\mathbf{x}). \tag{3.3}$$

Unlike the pair-wise design in DPO or IPO that cancels the log normalizing factor $\log Z_{\pi_t}(\mathbf{x})$ by differentiating (3.3) between $\mathbf{y}$ and $\mathbf{y}'$, we choose to approximate (3.3) directly in terms of $L_2$ distance:

$$\pi_{t+1} = \arg\min_{\pi} \mathbb{E}_{\mathbf{x} \sim \mathcal{X}, \mathbf{y} \sim \pi_t(\cdot|\mathbf{x})} \left( \log\left(\frac{\pi(\mathbf{y}|\mathbf{x})}{\pi_t(\mathbf{y}|\mathbf{x})}\right) - \big(\eta \mathbb{P}(\mathbf{y} \succ \pi_t|\mathbf{x}) - \log Z_{\pi_t}(\mathbf{x})\big) \right)^2. \tag{3.4}$$

**Estimation of the Probability** The optimization objective (3.4) can be approximated with finite samples. We choose to sample $K$ responses $\mathbf{y}_1, \mathbf{y}_2, \ldots, \mathbf{y}_K \sim \pi_t(\cdot|\mathbf{x})$ for each prompt $\mathbf{x}$, and denote the empirical distribution by $\widehat{\pi}_t^K$. The finite-sample optimization problem can be approximated as

$$\pi_{t+1} = \operatorname*{argmin}_{\pi} \mathbb{E}_{\mathbf{x} \sim \mathcal{X}, \mathbf{y} \sim \pi_t(\cdot|\mathbf{x})} \left( \log \left( \frac{\pi(\mathbf{y}|\mathbf{x})}{\pi_t(\mathbf{y}|\mathbf{x})} \right) - \left( \eta \mathbb{P}(\mathbf{y} \succ \widehat{\pi}_t^K|\mathbf{x}) - \log Z_{\widehat{\pi}_t^K}(\mathbf{x}) \right) \right)^2. \quad (3.5)$$

Specifically, $\mathbb{P}(\mathbf{y} \succ \widehat{\pi}_t^K|\mathbf{x}) = \sum_{k=1}^{K} \mathbb{P}(\mathbf{y} \succ \mathbf{y}_k|\mathbf{x})/K$ and $Z_{\widehat{\pi}_t^K}(\mathbf{x}) = \mathbb{E}_{\mathbf{y} \sim \pi_t(\cdot|\mathbf{x})}[\exp(\eta \mathbb{P}(\mathbf{y} \succ \widehat{\pi}_t^K|\mathbf{x}))]$. $Z_{\widehat{\pi}_t^K}(\mathbf{x})$, treated as an expectation, can be further estimated by $B$ new samples with in total $O(KB)$ queries of the preference oracle $\mathbb{P}$. (3.5) is an efficiently tractable optimization problem. Informally speaking, when $K \to \infty$, (3.5) will recover (3.4). We have the following guarantee on the convergence of (3.4):

**Theorem 3.1.** Assume the optimization problem (3.4) is realizable. Denote $\pi_t$ as the policy obtained via (3.4) and the mixture policy $\bar{\pi}_T = \frac{1}{T} \sum_{t=1}^{T} \pi_t$. By setting $\eta = \Theta(1/\sqrt{T})$, we have that

$$\max_{\pi} \left[ \mathbb{P}(\pi \succ \bar{\pi}_T) \right] - \min_{\pi} \left[ \mathbb{P}(\pi \prec \bar{\pi}_T) \right] = O(1/\sqrt{T}).$$

Theorem 3.1 characterizes the convergence rate of the average policy across the time horizon $T$ towards the Nash equilibrium, in terms of the duality gap. The proof is based on Theorem 1 in Freund & Schapire (1999) with slight modification. For completeness, we include the proof in Appendix F.

Alternatively, we can avoid estimating $\log Z_{\widehat{\pi}_t^K}(\mathbf{x})$ by replacing it with a constant based on the human preference model. The choice of the constant is discussed in detail in Appendix E. Here, we replace $\log Z_{\widehat{\pi}_t^K}(\mathbf{x})$ with $\eta/2^2$ in (3.5) to obtain a more clear objective:

$$\pi_{t+1} = \operatorname*{argmin}_{\pi} \mathbb{E}_{\mathbf{x} \sim \mathcal{X}, \mathbf{y} \sim \pi_t(\cdot|\mathbf{x})} \left( \log \left( \frac{\pi(\mathbf{y}|\mathbf{x})}{\pi_t(\mathbf{y}|\mathbf{x})} \right) - \eta \left( \mathbb{P}(\mathbf{y} \succ \widehat{\pi}_t^K|\mathbf{x}) - \frac{1}{2} \right) \right)^2. \quad (3.6)$$

Intuitively, if a tie occurs (i.e., $\mathbb{P}(\mathbf{y} \succ \widehat{\pi}_t^K|\mathbf{x}) = 1/2$), we prefer the model does not update weight at $\mathbf{y}$. If $\mathbf{y}$ wins over $\widehat{\pi}_t^K$ on average (i.e., $\mathbb{P}(\mathbf{y} \succ \widehat{\pi}_t^K|\mathbf{x}) > 1/2$), then we increase the probability density at $\mathbf{y}$ to employ the advantage of $\mathbf{y}$ over $\widehat{\pi}_t^K$. In our experiments, we choose to minimize the objective (3.6). In Appendix B, we provide a detailed comparison between (3.6) and other loss functions such as DPO and IPO.

## 3.2 THE SPPO ALGORITHM

Based on the aforementioned theoretical framework, we propose the *Self-Play Preference Optimization* algorithm in Algorithm 1.

---
**Algorithm 1** `Self-Play Preference Optimization (SPPO)`
---
1: **input**: base policy $\pi_{\boldsymbol{\theta}_1}$, preference oracle $\mathbb{P}$, learning rate $\eta$, number of generated samples $K$.
2: **for** $t = 1, 2, \ldots$ **do**
3:  Generate synthetic responses by sampling $\mathbf{x} \sim \mathcal{X}$ and $\mathbf{y}_{1:K} \sim \pi_t(\cdot|\mathbf{x})$.
4:  Annotate the win-rate $\mathbb{P}(\mathbf{y}_k \succ \mathbf{y}_{k'}|\mathbf{x}), \forall k, k' \in [K]$.
5:  Select responses from $\mathbf{y}_{1:K}$ to form dataset $\mathcal{D}_t = \{(\mathbf{x}_i, \mathbf{y}_i, \widehat{P}(\mathbf{y}_i \succ \pi_t|\mathbf{x}_i))\}_{i \in [N]}$.
6:  Optimize $\pi_{\boldsymbol{\theta}_{t+1}}$ according to (3.6):

$$\boldsymbol{\theta}_{t+1} \leftarrow \operatorname*{argmin}_{\boldsymbol{\theta}} \mathbb{E}_{(\mathbf{x}, \mathbf{y}, \widehat{P}(\mathbf{y} \succ \pi_t|\mathbf{x})) \sim \mathcal{D}_t} \left( \log \left( \frac{\pi_{\boldsymbol{\theta}}(\mathbf{y}|\mathbf{x})}{\pi_t(\mathbf{y}|\mathbf{x})} \right) - \eta \left( \widehat{P}(\mathbf{y} \succ \pi_t|\mathbf{x}) - \frac{1}{2} \right) \right)^2. \tag{3.7}$$

7: **end for**

---

In each iteration $t$, Algorithm 1 will first generate $K$ responses $\mathbf{y}_1, \mathbf{y}_2, \ldots, \mathbf{y}_K$ according to $\pi_t(\cdot|\mathbf{x})$ for each prompt $\mathbf{x}$ (Line 3). Then, the preference oracle $\mathbb{P}$ will be queried to calculate the win rate

---

[2]Assuming the winning probability between any given pair is either 1 or 0 with equal chance, when $K \to \infty$, we can show that indeed $Z_{\widehat{\pi}_t^K}(\mathbf{x}) \to e^{\eta/2}$. Also see Appendix E for a complete derivation.

among the $K$ responses (Line 4). At Line 5, certain criteria can be applied to determine which response should be kept in the constructed dataset $\mathcal{D}_t$ and construct the prompt-response-probability triplet $(\mathbf{x}, \mathbf{y}, \widehat{P}(\mathbf{y} \succ \pi_t | \mathbf{x}))$. We will discuss the design choices later in Section 4. One straightforward design choice is to include all $K$ responses into $\mathcal{D}_t$ and each $\widehat{P}(\mathbf{y}_i \succ \pi_t | \mathbf{x})$ is estimated by comparing $\mathbf{y}_i$ to all $K$ responses. In total, $O(K^2)$ queries will be made. Then the algorithm will optimize (3.6) on the dataset $\mathcal{D}_t$ (Line 6).

### 3.3 CONNECTION TO POLICY GRADIENT

While SPPO is derived from the iterative framework (Freund & Schapire, 1999) for two-player games, the square loss in the SPPO objective (3.4) provides an alternative interpretation for SPPO as a *semi-online* variant of policy gradient method due to its special loss form. The difference from standard policy gradient is that it collects samples from $\pi_{\boldsymbol{\theta}_t}$ at the start of iteration $t$, rather than perform on-policy sampling at each gradient step.

Consider a general reward function $r(\mathbf{y}; \mathbf{x})$, the RLHF problem (2.2) can be written as:

$$\max_{\boldsymbol{\theta}} J(\boldsymbol{\theta}) := \mathbb{E}_{\mathbf{x} \sim \mathcal{X}, \mathbf{y} \sim \pi_{\boldsymbol{\theta}}(\cdot|\mathbf{x})} \left[ r(\mathbf{y}; \mathbf{x}) - \eta^{-1} \log \frac{\pi_{\boldsymbol{\theta}}(\mathbf{y}|\mathbf{x})}{\pi_{\text{ref}}(\mathbf{y}|\mathbf{x})} \right]. \tag{3.8}$$

The policy gradient of the objective $J(\boldsymbol{\theta})$ is:

$$\nabla J(\boldsymbol{\theta}) = \mathbb{E}_{\mathbf{x} \sim \mathcal{X}, \mathbf{y} \sim \pi_{\boldsymbol{\theta}}(\cdot|\mathbf{x})} \left[ \left( r(\mathbf{y}; \mathbf{x}) - \eta^{-1} \log \frac{\pi_{\boldsymbol{\theta}}(\mathbf{y}|\mathbf{x})}{\pi_{\text{ref}}(\mathbf{y}|\mathbf{x})} - b(\mathbf{x}) \right) \nabla \log \pi_{\boldsymbol{\theta}}(\mathbf{y}|\mathbf{x}) \right] \tag{3.9}$$

$$= \frac{\eta}{2} \mathbb{E}_{\mathbf{x} \sim \mathcal{X}, \mathbf{y} \sim \pi_{\boldsymbol{\theta}}(\cdot|\mathbf{x})} \left[ -\nabla \left( r(\mathbf{y}; \mathbf{x}) - \eta^{-1} \log \frac{\pi_{\boldsymbol{\theta}}(\mathbf{y}|\mathbf{x})}{\pi_{\text{ref}}(\mathbf{y}|\mathbf{x})} - b(\mathbf{x}) \right)^2 \right], \tag{3.10}$$

where the first line follows the policy gradient theorem (Sutton et al., 1999) and the baseline $b(\mathbf{x})$ is an arbitrary constant relying only on $\mathbf{x}$ used for variance reduction[3]. Comparing the square loss (3.10) with the SPPO objective (3.4) (rewritten below):

$$\boldsymbol{\theta}_{t+1} = \operatorname*{argmin}_{\boldsymbol{\theta}} \mathbb{E}_{\mathbf{x} \sim \mathcal{X}, \mathbf{y} \sim \pi_{\boldsymbol{\theta}_t}(\cdot|\mathbf{x})} \left[ \left( \mathbb{P}(\mathbf{y} \succ \pi_{\boldsymbol{\theta}_t} | \mathbf{x}) - \eta^{-1} \log \left( \frac{\pi_{\boldsymbol{\theta}}(\mathbf{y}|\mathbf{x})}{\pi_{\boldsymbol{\theta}_t}(\mathbf{y}|\mathbf{x})} \right) - \eta^{-1} \log Z_{\pi_{\boldsymbol{\theta}_t}}(\mathbf{x}) \right)^2 \right],$$

one can see that the win rate $\mathbb{P}(\mathbf{y} \succ \pi_{\boldsymbol{\theta}_t} | \mathbf{x})$ is exactly the reward SPPO aims to maximize, and $\eta^{-1} \log Z_{\pi_{\boldsymbol{\theta}_t}}(\mathbf{x})$ is in fact the best possible baseline–the (soft) value function. When the value function is not available in practice, it can be replaced by any constant baseline to reduce the variance of the policy gradient. We choose $1/2$ as a good approximation to $\eta^{-1} \log Z_{\pi_{\boldsymbol{\theta}_t}}(\mathbf{x})$ but the constant can vary depending on the human preference model (see Appendix E).

Comparing with the general framework proposed by Swamy et al. (2024), SPPO can be seen as a new, straightforward variant of policy gradient method without the need of extra modifications such as gradient clipping in PPO, Hessian calculation in TRPO, or maintaining multiple components (Q-critic, V-critic, actor, etc.) in many policy optimization algorithms.

### 3.4 TOKEN-LEVEL $Q^*$ LEARNING

Rafailov et al. (2024a) showed that under the Max-Entropy RL formulation, the token-level log-ratio $\log \frac{\pi_{\boldsymbol{\theta}}(\mathbf{y}|\mathbf{x})}{\pi_{\text{ref}}(\mathbf{y}|\mathbf{x})}$ can be seen as an implicit token-level reward or advantage function (invariant under reward shaping). Below we show the square loss in SPPO can also lead to the optimal Max-Entropy policy $\pi^*$, with token-level optimal value/advantage function.

We first briefly restate the setting and results in Rafailov et al. (2024b). The token-level MDP defines the state $\mathbf{s}_h = (\mathbf{x}, y_1, y_2, \ldots, y_{h-1})$ as the prefix tokens, and the action $\mathbf{a}_h = y_h$ as the next token. An auto-regressive language model $\pi(\mathbf{y}|\mathbf{x})$ can be viewed as a token-level policy $\pi(\mathbf{a}_h|\mathbf{s}_h)$ and the transition kernel is known and deterministic because it only concatenates the next token to the prefix to form a new token sequence $\mathbf{s}_{h+1} = (\mathbf{x}, y_1, y_2, \ldots, y_h)$.

The Max-Entropy RL setting again considers the reverse-KL regularized reward maximization problem (2.2):

$$\max_{\boldsymbol{\theta}} \mathbb{E}_{\mathbf{x} \sim \mathcal{X}, \mathbf{y} \sim \pi_{\boldsymbol{\theta}}(\cdot|\mathbf{x})} [r(\mathbf{y}; \mathbf{x})] - \eta^{-1} \mathbb{E}_{\mathbf{x} \sim \mathcal{X}} [\text{KL}(\pi_{\boldsymbol{\theta}}(\cdot|\mathbf{x}) \| \pi_{\text{ref}}(\cdot|\mathbf{x}))]$$

---

[3]Equation (3.9) is also discussed in Munos et al. (2023). More recently, the same gradient form (3.10) has also been analyzed by Tang et al. (2025)

$$= \mathbb{E}_{\mathbf{x} \sim \mathcal{X}, \mathbf{y} \sim \pi_{\boldsymbol{\theta}}(\cdot|\mathbf{x})}[r(\mathbf{y};\mathbf{x}) + \eta^{-1} \log \pi_{\text{ref}}(\mathbf{y}|\mathbf{x})] + \eta^{-1} \mathbb{E}_{\mathbf{x} \sim \mathcal{X}}[\mathcal{H}(\pi_{\boldsymbol{\theta}}(\cdot|\mathbf{x}))].$$

We denote the optimal solution for the problem above as $\pi^*$. Rafailov et al. (2024a) showed that the Bradley-Terry preference model (B.2) can be rewritten as:

$$\mathbb{P}(\mathbf{y}_w \succ \mathbf{y}_l|\mathbf{x}) = \sigma\left(\eta^{-1} \sum_{h=1}^{|\mathbf{y}_w|} \log \frac{\pi^*(\mathbf{a}_h^w|\mathbf{s}_h^w)}{\pi_{\text{ref}}(\mathbf{a}_h^w|\mathbf{s}_h^w)} - \eta^{-1} \sum_{h=1}^{|\mathbf{y}_l|} \log \frac{\pi^*(\mathbf{a}_h^l|\mathbf{s}_h^l)}{\pi_{\text{ref}}(\mathbf{a}_h^l|\mathbf{s}_h^l)}\right),$$

where the state and action is defined as in the token-level MDP introduced above, with superscription $(\cdot)^w$ and $(\cdot)^l$ denoting if it is for the winner $\mathbf{y}_w$ or the loser $\mathbf{y}_l$. And maximizing the log likelihood with $\pi^*$ replaced by $\pi_{\boldsymbol{\theta}}$ gives the DPO loss.

From now on we assume the horizon is fixed at $H$ for simplicity. The derivation of the Max-Entropy RL formulation relies on the (soft) optimal value function $Q^*$ and $V^*$ as[4]:

$$V^*(\mathbf{s}_{H+1}) = r(\mathbf{s}_{H+1}) := r(\mathbf{y};\mathbf{x}), \text{ (reward at EOS)}$$
$$Q^*(\mathbf{s}_h, \mathbf{a}_h) = \eta^{-1} \log \pi_{\text{ref}}(\mathbf{a}_h|\mathbf{s}_h) + V^*(\mathbf{s}_{h+1}),$$
$$V^*(\mathbf{s}_h) = \eta^{-1} \log \sum_{\mathbf{a}} \exp\left(\eta Q^*(\mathbf{s}_h, \mathbf{a})\right), \text{ when } h \leq H.$$

Rafailov et al. (2024a) showed that the optimal policy $\pi^*$ satisfies:

$$\eta^{-1} \log \pi^*(\mathbf{a}_h|\mathbf{s}_h) = Q^*(\mathbf{s}_h, \mathbf{a}_h) - V^*(\mathbf{s}_h)$$
$$= \eta^{-1} \log \pi_{\text{ref}}(\mathbf{a}_h|\mathbf{s}_h) + V^*(\mathbf{s}_{h+1}) - V^*(\mathbf{s}_h).$$

It can be verified that for $\mathbf{s}_1 = (\mathbf{x})$, we have $\eta V^*(\mathbf{s}_1) = \log \sum_{\mathbf{y}} \pi_{\text{ref}}(\mathbf{y}|\mathbf{x}) \exp\left(\eta r(\mathbf{y};\mathbf{x})\right)$. Going back to the SPPO objective (3.4) at $t$-th iteration, if we set $\pi_{\text{ref}} = \pi_t$ and $r(\mathbf{y};\mathbf{x}) = \mathbb{P}(\mathbf{y} \succ \pi_t|\mathbf{x})$, we have $V^*(\mathbf{s}_1) = \eta^{-1} \log Z_{\pi_t}(\mathbf{x})$, and the learning objective at $t$-th iteration becomes:

$$\pi_{t+1} = \operatorname*{argmin}_{\pi} \mathbb{E}_{\mathbf{x} \sim \mathcal{X}, \mathbf{y} \sim \pi_t(\cdot|\mathbf{x})} \left( \log\left(\frac{\pi(\mathbf{y}|\mathbf{x})}{\pi_t(\mathbf{y}|\mathbf{x})}\right) - \left(\eta \mathbb{P}(\mathbf{y} \succ \pi_t|\mathbf{x}) - \log Z_{\pi_t}(\mathbf{x})\right) \right)^2$$
$$= \operatorname*{argmin}_{\pi} \mathbb{E}_{\mathbf{s}_1 \sim \mathcal{X}, \mathbf{a}_h \sim \pi_t(\cdot|\mathbf{s}_h)} \left( \sum_{h=1}^{H} \log \frac{\pi(\mathbf{a}_h|\mathbf{s}_h)}{\pi^*(\mathbf{a}_h|\mathbf{s}_h)} \right)^2. \tag{3.11}$$

Similar to DPO, SPPO "secretly" encourages the policy $\pi_{\boldsymbol{\theta}}$ to converge to the optimal policy $\pi^*$ at token level via the square loss form (3.11). Additionally, one may realize that minimizing the square-loss form is related to minimizing the KL divergence $\text{KL}(\pi_{\boldsymbol{\theta}} \| \pi^*)$ via policy gradient:

$$\nabla_{\boldsymbol{\theta}} \text{KL}(\pi_{\boldsymbol{\theta}} \| \pi^*) = \mathbb{E}_{\mathbf{s}_1 \sim \mathcal{X}, \mathbf{a}_h \sim \pi_{\boldsymbol{\theta}}(\cdot|\mathbf{s}_h)} \left[ \left( \sum_{h=1}^{H} \log \frac{\pi_{\boldsymbol{\theta}}(\mathbf{a}_h|\mathbf{s}_h)}{\pi^*(\mathbf{a}_h|\mathbf{s}_h)} \right) \sum_{h=1}^{H} \nabla_{\boldsymbol{\theta}} \log \pi_{\boldsymbol{\theta}}(\mathbf{a}_h|\mathbf{s}_h) \right]$$
$$= \mathbb{E}_{\mathbf{s}_1 \sim \mathcal{X}, \mathbf{a}_h \sim \pi_{\boldsymbol{\theta}}(\cdot|\mathbf{s}_h)} \left[ \nabla_{\boldsymbol{\theta}} \left( \sum_{h=1}^{H} \log \frac{\pi_{\boldsymbol{\theta}}(\mathbf{a}_h|\mathbf{s}_h)}{\pi^*(\mathbf{a}_h|\mathbf{s}_h)} \right)^2 \right].$$

# 4 EXPERIMENTS

## 4.1 EXPERIMENT SETUP

We briefly summarize our experiment setup as below. For a full description of our experiment setup, see Section C.

**Base Model and Datasets:** We follow Snorkel's experimental setup, using Mistral-7B-Instruct-v0.2 and Llama-3-8B-Instruct as our base models and Ultrafeedback for prompts. We split the dataset into three portions to avoid overfitting and ensure fair comparison with Snorkel.

**Preference Model:** We use PairRM, a 0.4B pair-wise preference model based on DeBERTA-V3, trained on high-quality human-preference datasets. PairRM outputs a "relative reward" between any given pair.

---

[4]Here we restated with the sequence-level reward $r(\mathbf{y};\mathbf{x})$. Rafailov et al. (2024a) started their derivation from a ground-truth token-level reward $r(\mathbf{s}_h, \mathbf{a}_h)$, which is under-specified due to the reward reshaping issue (Ng et al., 1999): reshaping the reward will not affect the Bradley-Terry preference probability so it is impossible to recover the ground-truth reward from the preference signal (Rafailov et al., 2024a, Section 4.2).

Table 1: AlpacaEval 2.0 evaluation of various models (detailed in Baselines) in terms of both normal and length-controlled (LC) win rates in percentage (%). Mistral-7B-SPPO Iter3 model achieves the highest LC win rate of 28.53% and a normal win rate of 31.02%. SPPO demonstrates steady performance gains across iterations and outperforms other baselines which show a tendency to produce longer responses. Additionally, re-ranking with the PairRM reward model (best-of-16) at test time consistently enhances the performance across all models and SPPO (best-of-16) achieves high win rate *without strong external supervision like GPT-4*. We additionally include the results obtained from fine-tuning Llama-3-8B-Instruct, which also show steady performance improvement.

| Model | AlpacaEval 2.0 | | |
| | LC Win Rate | Win Rate | Avg. Len |
| --- | --- | --- | --- |
| Mistral-7B-Instruct-v0.2 | 17.11 | 14.72 | 1676 |
| Mistral-7B-Instruct-v0.2 (best-of-16) | 22.45 | 17.94 | 1529 |
| Snorkel (Mistral-PairRM-DPO) | 26.39 | 30.22 | 2736 |
| Snorkel (Mistral-PairRM-DPO best-of-16) | 29.97 | 34.86 | 2616 |
| Self-Rewarding 70B Iter1 | - | 9.94 | 1092 |
| Self-Rewarding 70B Iter2 | - | 15.38 | 1552 |
| Self-Rewarding 70B Iter3 | - | 20.44 | 2552 |
| Mistral-7B-DPO Iter1 | 23.81 | 20.44 | 1723 |
| Mistral-7B-DPO Iter2 | 24.23 | 24.46 | 2028 |
| Mistral-7B-DPO Iter3 | 22.30 | 23.39 | 2189 |
| Mistral-7B-IPO Iter1 | 23.78 | 20.77 | 1693 |
| Mistral-7B-IPO Iter2 | 21.08 | 23.38 | 2660 |
| Mistral-7B-IPO Iter3 | 20.06 | 22.47 | 2760 |
| Mistral-7B-SPPO Iter1 | 24.79(+7.69) | 23.51(+8.79) | 1855 |
| Mistral-7B-SPPO Iter2 | 26.89(+2.10) | 27.62(+4.11) | 2019 |
| Mistral-7B-SPPO Iter3 | **28.53**(+1.64) | **31.02**(+3.40) | 2163 |
| Mistral-7B-SPPO Iter1 (best-of-16) | 28.71(+6.26) | 27.77(+9.83) | 1901 |
| Mistral-7B-SPPO Iter2 (best-of-16) | 31.23(+2.52) | 32.12(+4.35) | 2035 |
| Mistral-7B-SPPO Iter3 (best-of-16) | **32.13**(+0.9) | **34.94**(+2.82) | 2174 |
| Llama-3-8B-Instruct | 22.92 | 22.57 | 1899 |
| Llama-3-8B-SPPO Iter1 | 31.73(+8.81) | 31.74(+9.17) | 1962 |
| Llama-3-8B-SPPO Iter2 | 35.15(+3.42) | 35.98(+4.24) | 2021 |
| Llama-3-8B-SPPO Iter3 | **38.77**(+3.62) | **39.85**(+3.87) | 2066 |

**Response Generation and Selection:** We sample $K = 5$ responses per prompt with top $p = 1.0$ and temperature 1.0. We select the responses with the highest and lowest PairRM scores as the winning and losing responses respectively.

**Baselines and Benchmarks:** We evaluate Mistral-7B-Instruct-v0.2, Snorkel, iterative DPO and IPO for Mistral, and Self-rewarding LM. Benchmarks include AlpacaEval 2.0, MT-Bench, Arena-Hard, and the Open LLM Leaderboard. We also evaluate Llama-3-8B-Instruct on AlpacaEval 2.0.

## 4.2 EXPERIMENTAL RESULTS

**Evaluation using GPT-4 as a judge** Human evaluation remains the benchmark for quality and accuracy (Askell et al., 2021; Ouyang et al., 2022). However, due to its limitations in scalability and reproducibility, we explore the alternative approach of using the advanced capabilities of GPT-4 (OpenAI et al., 2023) as an automatic evaluation tool. We conduct GPT-4-based automatic evaluation on AlpacaEval 2.0 (Li et al., 2023b), MT-Bench (Zheng et al., 2023), and Arena-Hard (Li et al., 2024) to measure the chatbot capability of our model. The results can be found in Table 1 for AlpacaEval 2.0, Figure 3 (left) for MT-Bench, and Figure 3 (right) for Arena-Hard. We found that the performance of SPPO models consistently improves throughout all iterations (iterations).

Table 1 (AlpacaEval 2.0) shows the win rate over the GPT-4-Turbo baseline of different models on 805 prompts. We also include one column indicating the length-controlled win rate, and one column on the average length of each model, to account for the tendency of the LLM-based judge to favor longer sequence outputs — an issue colloquially termed the "reward hacking" phenomenon. According to the table, Mistral-7B-SPPO Iter3 has the highest win rate, 28.52% for the length-controlled version, and 31.02% for the overall win rate. The performance gains over previous iterations are 7.69% (Mistral-7B-Instruct → Iter1), 2.10% (Iter1 → Iter2), and 1.64% (Iter2 →

Table 2: AlpacaEval 2.0 leaderboard results of both normal and length-controlled (LC) win rates in percentage (%). Mistral-7B-SPPO can outperform larger models and Mistral-7B-SPPO (best-of-16) can outperform proprietary models such as GPT-4(6/13). Llama-3-8B-SPPO exhibits even better performance.

| Model | AlpacaEval 2.0 | |
|---|---|---|
| | LC. Win Rate | Win Rate |
| GPT-4 Turbo | 50.0 | 50.0 |
| Claude 3 Opus | 40.5 | 29.1 |
| Llama-3-8B-SPPO Iter3 | 38.8 | 39.9 |
| GPT-4 0314 | 35.3 | 22.1 |
| Llama 3 70B Instruct | 34.4 | 33.2 |
| Mistral-7B-SPPO Iter3 (best-of-16) | 32.1 | 34.9 |
| GPT-4 0613 | 30.2 | 15.8 |
| Snorkel (Mistral-PairRM-DPO best-of-16) | 30.0 | 34.9 |
| Mistral Medium | 28.6 | 21.9 |
| Mistral-7B-SPPO Iter3 | 28.5 | 31.0 |
| Claude 2 | 28.2 | 17.2 |
| Snorkel (Mistral-PairRM-DPO) | 26.4 | 30.2 |
| Gemini Pro | 24.4 | 18.2 |
| Mistral 8×7B v0.1 | 23.7 | 18.1 |
| Llama 3 8B Instruct | 22.9 | 22.6 |

| Model | MT-Bench | | |
|---|---|---|---|
| | 1st Turn | 2nd Turn | Average |
| Mistral-7B-Instruct-v0.2 | 7.78 | 7.25 | 7.51 |
| Snorkel (Mistral-PairRM-DPO) | 7.83 | 7.33 | 7.58 |
| Mistral-7B-DPO Iter1 | 7.45 | 6.58 | 7.02 |
| Mistral-7B-DPO Iter2 | 7.57 | 6.56 | 7.06 |
| Mistral-7B-DPO Iter3 | 7.49 | 6.69 | 7.09 |
| Mistral-7B-SPPO Iter1 | 7.63 | 6.79 | 7.21 |
| Mistral-7B-SPPO Iter2 | 7.90 | 7.08 | 7.49 |
| Mistral-7B-SPPO Iter3 | 7.84 | 7.34 | **7.59** |

| Model | Arena-Hard-Auto-v0.1 |
|---|---|
| Mistral-7B-Instruct | 12.6 |
| Snorkel (Mistral-PairRM-DPO) | 20.7 |
| Mistral-7B-SPPO Iter1 | 18.7 |
| Mistral-7B-SPPO Iter2 | 20.4 |
| Mistral-7B-SPPO Iter3 | **23.3** |

Table 3: **MT-Bench & Arena-Hard Evaluation.** Left: Mistral-7B-SPPO Iter3 outperforms all baseline models by achieving an average score of 7.59 in MT-Bench. Despite initial drops in performance in the first two iterations, SPPO Iter3 improves upon the base model by the final iteration. Right: Mistral-7B-SPPO Iter3 outperforms the baseline model Snorkel(Mistral-PairRM-DPO) in Arena-Hard. The improvement across different iterations is consistent.

Iter3), respectively, indicating steady improvements across iterations, as illustrated in Figure 1. We also apply SPPO to a stronger baseline model, i.e., Llama-3-8B-Instruct, and the fine-tuned model Llama-3-8B-SPPO has a higher length-controlled win rate 38.77% and overall win rate 39.85%. The performance gains are more significant: 8.81% (Llama-3-8B-Instruct → Iter1), 3.42% (Iter1 → Iter2), and 3.62% (Iter2 → Iter3), summing up to a total gain of 15.85%.

Additionally, the result indicates that SPPO achieves superior performance compared to the iterative variants of DPO and IPO. The length-controlled win rate for SPPO reaches 28.53%, outperforming the DPO's best rate of 26.39% (by Snorkel) and IPO's rate of 25.45%. Notably, while DPO and IPO training tend to significantly increase the average output length—2736 and 2654, respectively—SPPO shows a more moderate length increase, moving from 1676 in the base model to 2163 at the third iteration. Finally, we present the best-of-16 results for each model, selected using the PairRM reward model. We find that re-ranking with the preference model at test time can consistently improve the performance of base model (Mistral-7B-Instruct-v0.2), DPO (Snorkel), and SPPO (Iter3) by 5.34%, 3.57%, and 3.6%, respectively. Notably, this shows that while SPPO significantly enhances model alignment using PairRM-0.4B as the sole external supervision, it has not resulted in over-optimization against the preference model (Gao et al., 2023).

In Table 2, we compare SPPO on the AlpacaEval 2.0 leaderboard with other state-of-the-art AI chatbots. Our SPPO model outperforms many competing models trained on proprietary alignment data (e.g., Claude 2, Gemini Pro, & Llama 3 8B Instruct). When applied to Llama 3 8B Instruct, our Llama-3-8B-SPPO exhibits an even higher win rate. With test-time reranking, Mistral-7B-SPPO Iter3 (best-of-16) is even competitive to GPT-4 0613 and Llama 3 70B Instruct.

In Table 3 (left), we evaluate the performance of SPPO on MT-Bench. We can see that Mistral-7B-SPPO Iter3 outperforms all baseline models, achieving an average score of 7.59. While we are not certain why the MT-Bench performance drops at the first two iterations, the performance of SPPO at the final iteration still improves over the base model.

Table 4: **Open LLM Leaderboard Evaluation**. SPPO fine-tuning improves the base model's performance on different tasks, reaching a state-of-the-art average score of 66.75 for Mistral-7B and 70.29 for Llama-3-8B. For Mistral-7B, subsequent iterations of DPO, IPO, and SPPO see a decline in performance. It is possible that aligning with human preferences (simulated by the PairRM preference model in our study) may not always enhance, and can even detract from, overall performance.

| Models | Arc | TruthfulQA | WinoGrande | GSM8k | HellaSwag | MMLU | Average |
|---|---|---|---|---|---|---|---|
| Mistral-7B-Instruct-v0.2 | 63.65 | 66.85 | 77.98 | 41.93 | 84.89 | 59.15 | 65.74 |
| Snorkel | 66.04 | 70.86 | 77.74 | 36.77 | 85.64 | 60.83 | 66.31 |
| Mistral-7B-DPO Iter1 | 63.14 | 68.39 | 77.19 | 40.33 | 85.25 | 59.41 | 65.62 |
| Mistral-7B-DPO Iter2 | 64.16 | 67.84 | 76.09 | 39.95 | 85.23 | 59.03 | 65.38 |
| Mistral-7B-DPO Iter3 | 65.19 | 67.89 | 77.27 | 32.30 | 85.49 | 59.00 | 64.52 |
| Mistral-7B-IPO Iter1 | 64.68 | 68.60 | 77.98 | 43.75 | 85.08 | 59.04 | 66.52 |
| Mistral-7B-IPO Iter2 | 62.12 | 66.30 | 77.51 | 39.20 | 83.15 | 59.70 | 64.66 |
| Mistral-7B-IPO Iter3 | 62.97 | 67.12 | 77.51 | 37.45 | 83.69 | 59.57 | 64.72 |
| Mistral-7B-SPPO Iter1 | 65.02 | 69.40 | 77.82 | 43.82 | 85.11 | 58.84 | 66.67 |
| Mistral-7B-SPPO Iter2 | 65.53 | 69.55 | 77.03 | 44.35 | 85.29 | 58.72 | **66.75** |
| Mistral-7B-SPPO Iter3 | 65.36 | 69.97 | 76.80 | 42.68 | 85.16 | 58.45 | 66.40 |
| Llama-3-8B-Instruct | 62.29 | 51.65 | 76.09 | 75.89 | 78.73 | 65.59 | 68.37 |
| Llama-3-8B-SPPO Iter1 | 63.82 | 54.96 | 76.40 | 75.44 | 79.80 | 65.65 | 69.35 |
| Llama-3-8B-SPPO Iter2 | 64.93 | 56.48 | 76.87 | 75.13 | 80.39 | 65.67 | 69.91 |
| Llama-3-8B-SPPO Iter3 | 65.19 | 58.04 | 77.11 | 74.91 | 80.86 | 65.60 | **70.29** |

Arena-Hard (Li et al., 2024) contains 500 challenging user queries and follows the same evaluation method as AlpacaEval 2.0. In Table 3 (right), we evaluate the performance of SPPO on Arena-Hard. We can see that Mistral-7B-SPPO exhibits a steady performance gain across iterations. Mistral-7B-SPPO Iter 3 outperforms the baseline models, achieving an average score of 23.3.

**Open LLM Leaderboard**   We further evaluate the capabilities of SPPO models using Huggingface Open LLM Leaderboard (Beeching et al., 2023a). This leaderboard encompasses 6 different datasets, each focusing on a specific capability of LLMs: Arc (Clark et al., 2018), HellaSwag (Zellers et al., 2019), Winogrande (Sakaguchi et al., 2021), MMLU (Hendrycks et al., 2020), TruthfulQA (Lin et al., 2021), and GSM8k (Cobbe et al., 2021). The models are prompted with zero or few-shot exemplars. The results, presented in Table 4, demonstrate that SPPO can enhance the performance of the base model on Arc, TruthfulQA, and GSM8k, and achieve the state-of-the-art performance with an average score of 66.75. However, these improvements do not hold in subsequent alignment iterations: DPO, IPO, and SPPO's performance declines after the first or second iterations. This limitation may be attributed to the "alignment tax" phenomenon (Askell et al., 2021), which suggests that aligning with human preferences (simulated by PairRM preference in our study) might not improve or even hurt the general performance. Improving language model capabilities through alignment iterations remains a topic for future research, and we posit that incorporating high-quality SFT annotations (Chen et al., 2024) could play a significant role in this endeavor.

## 5   CONCLUSIONS

This paper introduced Self-Play Preference Optimization (SPPO), an approach to fine-tuning Large Language Models (LLMs) from Human/AI Feedback. SPPO has demonstrated significant improvements over existing methods such as DPO and IPO across multiple benchmarks, including AlpacaEval 2.0, MT-Bench, Arena-Hard, and the Open LLM Leaderboard. By integrating a preference model and employing a new optimization objective, SPPO can align LLMs more closely with human preferences.

**Limitations** Theoretically, approximating the optimal policy update via regression relies on the assumption that the model class is expressive enough and the generated data well cover the input space. Approximating the log-partition factor with a constant can help reduce variance only when it is close to the soft value function. The experiments are run on one dataset UltraFeedback and the models are tested on a few benchmarks due to limited computational resources, but the proposed methods can be further validated on more models, datasets, and benchmarks to have a holistic evaluation given more resources.

ACKNOWLEDGMENTS

We thank the anonymous reviewers and area chair for their helpful comments. YW, KJ and QG are supported in part by the NSF grants IIS-2008981, DMS-2323113, CPS-2312094, IIS-2403400 and Sloan Research Fellowship. YW is also supported by UCLA Dissertation Year Fellowship. The views and conclusions contained in this paper are those of the authors and should not be interpreted as representing any funding agencies.

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

## A RELATED WORK

**RLHF with Explicit/Implicit Reward Model** Originally, reinforcement learning from human feedback (RLHF) was proposed by Christiano et al. (2017) as a methodology that first learns a reward model reflecting human preferences and then uses reinforcement learning algorithms to maximize the reward. This methodology is applied by Ouyang et al. (2022) to fine-tune instruction-following large language models and leads to the popular ChatGPT.

The reward model in the works mentioned above assumes a parametric model such as the Bradley-Terry model (Bradley & Terry, 1952), which assigns a "score" representing how preferred a given response is. More recently, Rafailov et al. (2024b) proposed to instead directly solve the closed-form solution of such a score implied by the Bradley-Terry model. The Direct Policy Optimization (DPO) method is claimed to be more efficient and stable, yet, still implicitly assumes such a reward model that specifies the "score". In a similar spirit, Zhao et al. (2023) proposed to calibrate the score so that the score of the winner in comparison has a margin over the score of the loser, and induces a different SLic loss. Similarly, Ethayarajh et al. (2024) derived a different loss function (called KTO) from the Kahneman-Tversky human utility function, which implicitly denotes a score of the given response. Liu et al. (2023) proposed Rejection Sampling Optimization (RSO) which utilizes a preference model to generate preference pairs with candidates sampled from the optimal policy; then preference optimization is applied on the sampled preference pairs. Hong et al. (2024) proposed Odds Ratio Preference Optimization (ORPO) algorithm that can perform supervised fine-tuning and preference alignment in one training session without maintaining an intermediate reference policy. Motivated by the length bias in the preference dataset, Meng et al. (2024) model human preference

based on the average-reward Bradley-Terry model, and propose Simple Preference Optimization (SimPO) which maximizes the gap between the average log-likelihood of the winner and loser.

As discussed in Section 3.3, SPPO can be seen as a semi-online REINFORCE-style algorithm. The effectiveness of REINFORCE, rather than PPO with value function and gradient clipping, has been shown by Ahmadian et al. (2024), where they argue that REINFORCE should suffice for RLHF tasks. Our work also corroborates this observation.

**RLHF with General Preference Model**   Often, the human preference is not strictly transitive, and cannot be sufficiently represented by a single numerical score. Azar et al. (2023) proposed a general preference optimization objective based on the preference probability between a pair of responses instead of a score of a single response. They further propose a learning objective based on identity mapping of the preference probability called IPO (Preference Optimization with Identity mapping), which aims to maximize the current policy's expected winning probability over a given reference policy. Munos et al. (2023) formulated the RLHF problem with general preference as a two-player, constant-sum game, where each player is one policy that aims to maximize the probability of its response being preferred against its opponent. They aim to identify the Nash equilibrium policy of this game and propose a mirror-descent algorithm that guarantees the last-iterate convergence of a policy with tabular representations[5]. Wang et al. (2024) proposed to identify the Nash equilibrium policy for multi-step MDPs when a general preference model is present and shows that the problem can be reduced to a two-player zero-sum Markov game.

**Theory of RLHF**   There is also a line of research to analyze RLHF and provide its theoretical guarantees. Zhu et al. (2023) studied the standard RLHF with separate reward-learning and model-tuning and proposed a pessimistic reward-learning process that provably learns a linear reward model. Wang et al. (2024) proposed a framework to reduce any RLHF problem with a reward model to a reward-based standard RL problem. Additionally, they proposed to identify the Nash equilibrium policy when a general preference model is present and show that the problem can be reduced to a two-player zero-sum Markov game. Xiong et al. (2023) studied the reverse-KL regularized contextual bandit for RLHF in different settings and proposed efficient algorithms with finite-sample theoretical guarantees. Ye et al. (2024) studied the theoretical learnability of the KL-regularized Nash-Learning from Human Feedback (NLHF) by considering both offline and online settings and proposed provably efficient algorithms. Ji et al. (2024) proposed an active-query-based proximal policy optimization algorithm with regret bounds and query complexity based on the problem dimension and the sub-optimality gap.

**Self-Play Fine-Tuning**   Most works mentioned above (Rafailov et al., 2024b; Zhao et al., 2023; Azar et al., 2023; Ethayarajh et al., 2024) consider one single optimization procedure starting from some reference policy. The same procedure may be applied repeatedly for multiple iterations in a self-play manner. In each iteration, new data are generated by the policy obtained in the last iteration; these new data are then used for training a new policy that can outperform the old policy.

The self-play fine-tuning can be applied to both scenarios with or without human preference data. For example, Singh et al. (2023) proposed an Expectation-Maximization (EM) framework where in each iteration, new data are generated and annotated with a reward score; the new policy is obtained by fine-tuning the policy on the data with a high reward. Chen et al. (2024) proposed a self-play framework to fine-tune the model in a supervised way. In each iteration, new preference pairs are synthesized by labeling the policy-generated responses as losers and the human-generated responses as winners. Then DPO is applied in each iteration to fine-tune another policy based on these synthesized preference data. Yuan et al. (2024) proposed Self-Rewarding Language Models, where the language model itself is used to annotate preference on its own responses. Iterative DPO is applied to fine-tune language models on these annotated data. These works show iterative fine-tuning can significantly improve the performance.

Munos et al. (2023) are among the first to introduce self-play algorithms for aligning large language models with human preferences by computing Nash equilibria of a preference model rather than optimizing a reward model. Their proposed method, Nash-MD, based on mirror descent, provides a sequence of policies with the final iteration converging to the regularized Nash equilibrium. However, Nash-MD requires exact sampling from the mixture distribution that is only feasible in the

---

[5]Due to the tabular representation, computing the normalizing factor is prohibitive and the algorithm is approximately executed by sampling one token instead of a full response.

bandit setting, while sequential generation necessitates an approximation that may impact theoretical guarantees.

Swamy et al. (2024) considered a more general multi-step Markov Decision Process (MDP) setting and proposed Self-play Preference Optimization (SPO), an RLHF framework that can utilize any no-regret online learning algorithm for preference-based policy optimization. They then instantiated their framework with Soft Policy Iteration as an idealized variant of their algorithm, which reduces to the exponential weight update rule (3.1) when constrained to the bandit setting. The main difference is that they focus on the multi-round Markov decision process (MDP) in robotic and game tasks rather than on fine-tuning large language models and approximating the update using policy optimization methods such as PPO.

Concurrent to our work, Rosset et al. (2024) proposed the Direct Nash Optimization (DNO) algorithm based on the cross-entropy between the true and predicted win rate gaps, and provided theoretical guarantees on the error of finite-sample approximation. However, their practical version still utilizes the iterative-DPO framework as in Xu et al. (2023) with the DPO loss instead of their derived DNO loss. Notably, in their experiments, they added the GPT-4 generated responses as their "gold sample" into their fine-tuning data, and used GPT-4 as a judge to assign a numerical score to each response for preference pair construction. In sharp contrast, our work does not require the use of any strong external supervision besides a small-sized reward model. Another concurrent work (Gao et al., 2024) proposed REBEL, an iterative fine-tuning framework via regressing the relative reward. When applied to the preference setting, it results in a similar algorithm to our algorithm SPPO, except that SPPO approximates the log-partition factor $\log Z_{\pi_t}(\mathbf{x})$ with a constant $\eta/2$ while REBEL regresses on the win rate difference (so that $\log Z_{\pi_t}(\mathbf{x})$ is canceled). Additionally, Calandriello et al. (2024) pointed out that optimizing the IPO loss (Azar et al., 2023) iteratively with self-play generated data is equivalent to finding the Nash equilibrium of the two-player game, and they proposed the IPO-MD algorithm based on this observation, which generates data with a mixture policy similar to the Nash-MD algorithm.

## B  COMPARISON WITH DPO, IPO, AND KTO

In practice, we utilize mini-batches of more than 2 responses to estimate the win rate of a given response, while the DPO and IPO loss focus on a single pair of responses. When only a pair of responses $\mathbf{y}_w$ and $\mathbf{y}_l$ is available, we have the pair-wise symmetric loss based on the preference triplet $(\mathbf{x}, \mathbf{y}_w, \mathbf{y}_l)$ defined as:

$$\ell_{\mathrm{SPPO}}(\mathbf{x}, \mathbf{y}_w, \mathbf{y}_l; \boldsymbol{\theta}; \pi_{\mathrm{ref}}) := \left( \log\left( \frac{\pi_{\boldsymbol{\theta}}(\mathbf{y}_w|\mathbf{x})}{\pi_{\mathrm{ref}}(\mathbf{y}_w|\mathbf{x})} \right) - \eta\left( \mathbb{P}(\mathbf{y}_w \succ \mathbf{y}_l|\mathbf{x}) - \frac{1}{2} \right) \right)^2$$
$$+ \left( \log\left( \frac{\pi_{\boldsymbol{\theta}}(\mathbf{y}_l|\mathbf{x})}{\pi_{\mathrm{ref}}(\mathbf{y}_l|\mathbf{x})} \right) - \eta\left( \mathbb{P}(\mathbf{y}_w \prec \mathbf{y}_l|\mathbf{x}) - \frac{1}{2} \right) \right)^2, \quad \text{(B.1)}$$

where $\mathbb{P}(\mathbf{y}_w \succ \mathbf{y}_l|\mathbf{x})$ can be either a soft probability within $[0,1]$ or a hard label 1 indicating $\mathbf{y}_w \succ \mathbf{y}_l$.

We now compare the SPPO loss to other baselines assuming a hard label $\mathbf{y}_w \succ \mathbf{y}_l$ is given. For the ease of comparison, let $(\beta = \eta^{-1})$:

$$a = \beta \log\left( \frac{\pi_{\boldsymbol{\theta}}(\mathbf{y}_w|\mathbf{x})}{\pi_{\mathrm{ref}}(\mathbf{y}_w|\mathbf{x})} \right), b = \beta \log\left( \frac{\pi_{\boldsymbol{\theta}}(\mathbf{y}_l|\mathbf{x})}{\pi_{\mathrm{ref}}(\mathbf{y}_l|\mathbf{x})} \right), c = \beta \mathrm{KL}(\pi_{\boldsymbol{\theta}} \| \pi_{\mathrm{ref}}),$$

then we have

$$\ell_{\mathrm{DPO}}(\mathbf{y}_w, \mathbf{y}_l, \mathbf{x}) = -\log \sigma(a - b), \quad \text{(B.2)}$$
$$\ell_{\mathrm{IPO}}(\mathbf{y}_w, \mathbf{y}_l, \mathbf{x}) = [(a - b) - 1]^2, \quad \text{(B.3)}$$
$$\ell_{\mathrm{KTO}}(\mathbf{y}_w, \mathbf{y}_l, \mathbf{x}) = \sigma(-a + c) + \sigma(b - c) \text{ (simplified)}, \quad \text{(B.4)}$$

where $\sigma(x) = e^x/(e^x + 1)$ and the SPPO loss can be written as

$$\ell_{\mathrm{SPPO}}(\mathbf{y}_w, \mathbf{y}_l, \mathbf{x}) = (a - 1/2)^2 + (b + 1/2)^2.$$

It can be seen that SPPO not only pushes the gap between $a$ and $b$ to be 1, but also attempts to push value of $a$ to be close to $1/2$ and the value of $b$ to be close to $-1/2$ such that $\pi_{\boldsymbol{\theta}}(\mathbf{y}_w|\mathbf{x}) > \pi_{\mathrm{ref}}(\mathbf{y}_w|\mathbf{x})$ and $\pi_{\boldsymbol{\theta}}(\mathbf{y}_l|\mathbf{x}) < \pi_{\mathrm{ref}}(\mathbf{y}_l|\mathbf{x})$. We believe this is particularly important: when there are plenty of

preference pairs, DPO and IPO can ensure the policy will converge to the target policy, but when the preference pairs are scarce (e.g., one pair for each prompt), there is no guarantee that the estimated reward of the winner $a$ will increase and the estimated reward of the loser $b$ will decrease. Instead, only the reward gap between the winner and the loser (i.e., $a - b$) will increase. This phenomenon is observed by Pal et al. (2024) that DPO only drives the loser's likelihood to be small, but the winner's likelihood barely changes. We believe that fitting $\beta \log \left( \frac{\pi_{t+1}(\mathbf{y}|\mathbf{x})}{\pi_t(\mathbf{y}|\mathbf{x})} \right)$ directly to $\mathbb{P}(\mathbf{y} \succ \pi_t | \mathbf{x}) - 1/2$ is more effective than IPO which attempts to fit $\beta \log \left( \frac{\pi_{t+1}(\mathbf{y}_w|\mathbf{x})}{\pi_t(\mathbf{y}_w|\mathbf{x})} \right) - \beta \log \left( \frac{\pi_{t+1}(\mathbf{y}_l|\mathbf{x})}{\pi_t(\mathbf{y}_l|\mathbf{x})} \right)$ to $\mathbb{P}(\mathbf{y}_w \succ \pi_t | \mathbf{x}) - \mathbb{P}(\mathbf{y}_l \succ \pi_t | \mathbf{x})$. In addition, SPPO shares a similar spirit as KTO. The KTO loss pushes $a$ to be large by minimizing $\sigma(-a + c)$ and pushes $b$ to be small by minimizing $\sigma(b - c)$. In contrast, SPPO pushes $a$ to be as large as $1/2$ and $b$ to be as small as $-1/2$.

On the other hand, we would like to comment that although DPO and KTO can be extended to their iterative variants, they are not by nature iterative algorithms and do not have provable guarantees that they can reach the Nash equilibrium. In contrast, SPPO and IPO are by design capable to solve the Nash equilibrium iteratively. SPPO is superior to IPO because its design explicitly alleviates the data sparsity issue, as discussed above and detailed in Pal et al. (2024).

## C FULL EXPERIMENT SETUP

**Base Model and Datasets**   We follow the experimental setup of Snorkel[6], a model that utilizes iterative DPO to achieve state-of-the-art performance on AlpacaEval benchmarks. Specifically, we use Mistral-7B-Instruct-v0.2 as our base model[7]. Mistral-7B-Instruct-v0.2 is an instruction fine-tuned version of Mistral-7B-v0.2 model (Jiang et al., 2023a). We also adopt Ultrafeedback (Cui et al., 2023) as our source of prompts which includes around 60k prompts from diverse resources. During generation, we follow the standard chat template of Mistral-7B. To avoid overfitting during the fine-tuning, we split the dataset into three portions and used only one portion per iteration. These settings were also adopted by training the model Snorkel-Mistral-PairRM-DPO[8] (Snorkel). We follow the splitting in Snorkel for a fair comparison. Additionally, we use Llama-3-8B-Instruct[9] as a stronger base model along with the same preference dataset and data splitting.

**Preference Model**   We employ PairRM (Jiang et al., 2023b), an efficient pair-wise preference model of size 0.4B. PairRM is based on DeBERTA-V3 (He et al., 2021) and trained on high-quality human-preference datasets. Results on benchmarks like Auto-J Pairwise dataset (Li et al., 2023a) show that it outperforms most of the language-model-based reward models and performs comparably with larger reward models like UltraRM-13B (Cui et al., 2023). We refer the readers to the homepage on Huggingface[10] for detailed benchmark results. We therefore keep PairRM as our ranking model following Snorkel for a balance between accuracy and efficiency.

Specifically, PairRM will output a "relative reward" $s(\mathbf{y}, \mathbf{y}'; \mathbf{x})$ that reflects the strength difference between $\mathbf{y}$ and $\mathbf{y}'$, i.e.,

$$\mathbb{P}(\mathbf{y} \succ \mathbf{y}'|\mathbf{x}) = \frac{\exp(s(\mathbf{y}, \mathbf{y}'; \mathbf{x}))}{1 + \exp(s(\mathbf{y}, \mathbf{y}'; \mathbf{x}))}.$$

Unlike the Bradley-Terry-based reward model, PairRM only assigns the relative reward which is not guaranteed to be transitive (i.e., $s(\mathbf{y}_1, \mathbf{y}_2; \mathbf{x}) + s(\mathbf{y}_2, \mathbf{y}_3; \mathbf{x}) \neq s(\mathbf{y}_1, \mathbf{y}_3; \mathbf{x})$). So it indeed models the general preference.

**Response Generation and Selection**   During the generation phase in each iteration, we use top $p = 1.0$ and temperature 1.0 to sample from the current policy. We sample with different random seeds to get $K = 5$ different responses for each prompt. Previous works utilizing Iterative DPO choose 2 responses to form a pair for each prompt. For a fair comparison, we do not include all $K = 5$ responses in the preference data but choose two responses among them. Following Snorkel, we choose the winner $\mathbf{y}_w$ and loser $\mathbf{y}_l$ to be the response with the *highest* and *lowest* PairRM score,

---

which is defined for each response $\mathbf{y}_i$ as:

$$s_{\text{PairRM}}(\mathbf{y}_i; \mathbf{x}) := \frac{1}{K} \sum_{k=1}^{K} s(\mathbf{y}_i, \mathbf{y}_k; \mathbf{x}).$$

**Probability Estimation** We then estimate the win rate over the distribution by the average win rate over all the sampled responses as explained in (3.5):

$$\widehat{P}(\mathbf{y}_i \succ \pi_t | \mathbf{x}) = \frac{1}{K} \sum_{k=1}^{K} \mathbb{P}(\mathbf{y}_i \succ \mathbf{y}_k | \mathbf{x}), \forall i \in [K].$$

**Hyperparameter Tuning** The experiments are conducted on $8 \times$ Nvidia A100 GPUs. For SPPO, we trained three iterations in total. In each iteration, we selected the model trained on the first epoch of the 20k prompts from UltraFeedback to proceed to the next iteration. For both Mistral-7B-Instruct-v0.2 and Llama-3-8B-Instruct, the global training batch size is set to 64, and $\eta$ is set to $1e3$. The learning rate schedule is determined by the following hyperparameters: learning rate=5.0e-7, number of total training epochs=18, warmup ratio=0.1, linear schedule. In practice, early stopping after the first epoch yields the best test performance. The best hyper-parameters for each model are selected by the average win rate (judged by PairRM-0.4B) on a hold-out subset of Ultrafeedback as the metric. For more details on the win-rate comparison using PairRM as a judge, please refer to Section 4.2 and Figure 2.

**Baselines** We evaluate the following base models as well as baseline methods for fine-tuning LLMs:

- Mistral-7B-Instruct-v0.2: Mistral-7B-Instruct-v0.2 is an instruction fine-tuned version of Mistral-7B-v0.2 model (Jiang et al., 2023a). It is the starting point of our algorithm.

- Snorkel (Mistral-PairRM-DPO): We directly evaluate the uploaded checkpoint on HuggingFace[11]. This model is obtained by three iterations of iterative DPO from Mistral-7B-Instruct-v0.2.

- (Iterative) DPO: We also implement the iterative DPO algorithm by ourselves. The experimental settings and model selection schemes align with those used for SPPO, except for the adoption of the DPO loss function as defined in (B.2). Hyperparameters are optimized to maximize the average win-rate assessed by PairRM at each iteration. Note that the practical algorithm in Rosset et al. (2024) is essentially the same as iterative DPO.

- (Iterative) IPO: We implement the iterative IPO algorithm by ourselves. The experimental setting and the model selection scheme is the same as iterative DPO, except that the loss function is the IPO loss (B.3). For fair comparison, hyperparameters for IPO is also selected by evaluation using the average PairRM win-rate on the hold-out subset of Ultrafeedback.

- Self-rewarding LM: Yuan et al. (2024) proposed to prompt the LLM itself as a preference judge to construct new preference pairs and iteratively fine-tune the LLM with the DPO algorithm. We use the AlpacaEval 2.0 win rate reported by Yuan et al. (2024) for comparison. Note that Self-rewarding LM is a trained from Llama 2 70B.

- Llama-3-8B-Instruct: Llama-3-8B-Instruct is an instruction-tuned model optimized for dialogue use cases and outperforms many of the available open-source chat models on common industry benchmarks.

**Benchmarks** Following previous works, we use AlpacaEval 2.0 (Dubois et al., 2024a), Arena-Hard(Li et al., 2024), MT-Bench (Zheng et al., 2024), and Open LLM Leaderboard (Beeching et al., 2023b) as our evaluation benchmarks.

- **AlpacaEval 2.0** is an LLM-based automatic evaluation benchmark. It employs AlpacaFarm (Dubois et al., 2024b) as its prompts set composed of general human instructions. The model responses and the reference response generated by GPT-4-Turbo are fed into a GPT-4-Turbo-based annotator to be judged. We follow the standard approach and report the win rate over the reference responses.

- **Arena-Hard** (Li et al., 2024) is a high-quality benchmark that claims to be harder and has the highest correlation and separability to Chatbot Arena among popular open-ended LLM benchmarks including AlpacaEval 2.0. We evaluate our models Mistral-PairRM-SPPO and the baseline models.

---

[11]https://huggingface.co/snorkelai/Snorkel-Mistral-PairRM-DPO

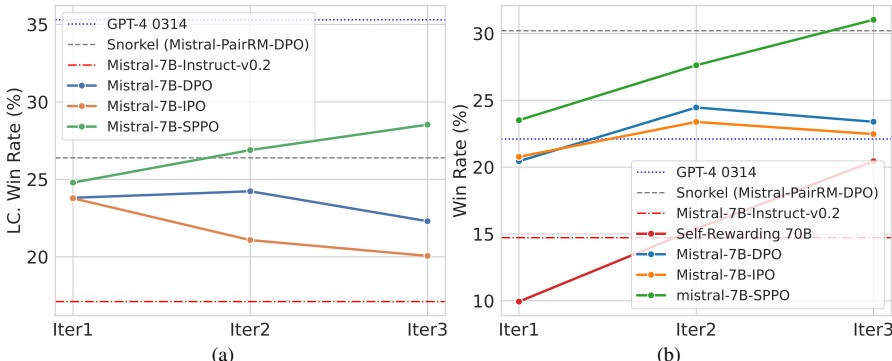

Figure 1: Win Rate against GPT-4-Turbo with (a) and without (b) Length Controlling (LC) on AlpacaEval 2.0. SPPO demonstrates steady improvements on both LC and raw win rates.

- **MT-Bench** (Zheng et al., 2024) is a collection of 80 high-quality multi-turn open-ended questions. The questions cover topics like writing, role-playing, math, coding, etc.. The generated answer is judged by GPT-4 and given a score directly without pairwise comparison.

- **Open LLM Leaderboard** (Beeching et al., 2023b) consists of six datasets, each of which focuses on a facet of language model evaluation. In detail, the evaluation rubric includes math problem-solving, language understanding, human falsehood mimicking, and reasoning. We follow the standard evaluation process and use in-context learning to prompt the language model and compute the average score over six datasets to measure the performance.

# D    ADDITIONAL EXPERIMENT RESULTS

## D.1    ADDITIONAL TABLES AND PLOTS

In Figure 1, we plot the win rate against GPT-4-Turbo on AlpacaEval 2.0 of different RLHF algorithms. We can see that the performance gains of SPPO over previous iterations are 7.69% (Mistral-7B-Instruct → Iter1), 2.10% (Iter1 → Iter2), and 1.64% (Iter2 → Iter3), respectively, indicating steady improvements across iterations.

## D.2    EVALUATION USING PAIRRM AS A JUDGE

As SPPO identifies the von Neumann winner (see (2.3)) in a two-player constant-sum game, we examine the pairwise preferences among SPPO models and other baselines. The pairwise win rates, measured by PairRM, are depicted in Figure 2. We observe that in all algorithms—namely DPO, IPO, and SPPO—the newer model iterations surpass the previous ones. For example, SPPO iteration 3 outperforms SPPO iteration 2. Both SPPO and IPO consistently outperform DPO across all iterations. While SPPO is superior to IPO in the first two iterations, IPO exceeds SPPO in performance during the final iteration. Considering the superior performance of SPPO in standard benchmarks evaluated by GPT-4 or against ground-truth answers (e.g., AlpacaEval 2.0, MT-Bench, and Open LLM Leaderboard), along with IPO's tendency to produce longer sequence outputs (see Avg. Len in Table 1), we believe this is due to IPO exploiting the length bias in PairRM that favors longer sequences. Conversely, SPPO models benefit from a more robust regularization within a multiplicative weight update framework.

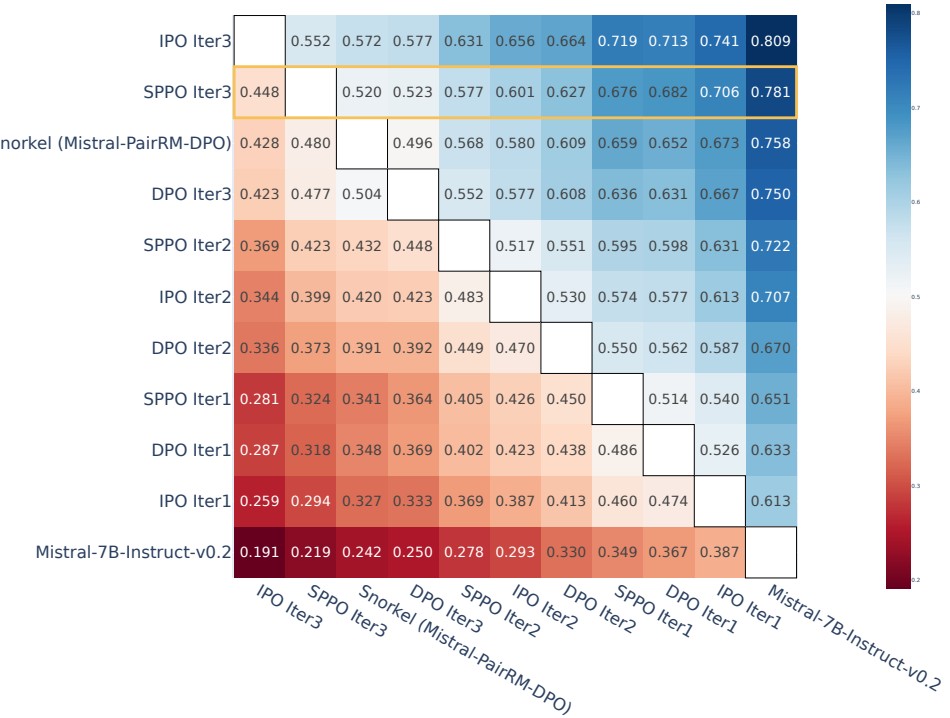

Figure 2: Pairwise win rates among base model (Mistral-7B-Instruct-v0.2), DPO models, IPO models, and SPPO models using **PairRM-0.4B** as a judge, which may favor models with longer outputs. On benchmarks with more powerful judge models (e.g., GPT-4), such as AlpacaEval 2.0 and MT-Bench, SPPO outperforms other baseline algorithms by a large margin.

## D.3 ABLATION STUDY

| Mini-Batch Size | Iteration | AlpacaEval 2.0 Win Rate LC. | Raw | Avg. Len (chars) |
|---|---|---|---|---|
| $K = 2$ | Iter1 | 23.85 | 23.53 | 1948 |
| | Iter2 | 26.91 | 27.24 | 1999 |
| | Iter3 | 28.26 | 28.22 | 1961 |
| $K = 5$ | Iter1 | 24.79 | 23.51 | 1855 |
| | Iter2 | 26.89 | 27.62 | 2019 |
| | Iter3 | **28.53** | **31.02** | 2163 |

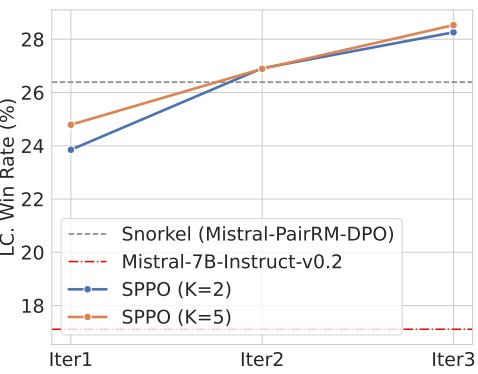

Figure 3: AlpacaEval 2.0 evaluation on SPPO of different mini-batch size in terms of both normal and length-controlled (LC) win rates in percentage (%). $K = 2, 5$ denote different mini-batch sizes when estimating the win rate $\mathbb{P}(\mathbf{y} \succ \pi_t | \mathbf{x})$.

We study the effect of mini-batch size when estimating the win rate $\mathbb{P}(\mathbf{y} \succ \pi_t | \mathbf{x})$. Specifically, for each prompt, we still generate 5 responses and choose the winner $\mathbf{y}_w$ and loser $\mathbf{y}_l$ according to the PairRM score. When estimating the probability, we vary the batch size to be $K = 2, 3, 5$. For $K = 2$, we estimate $\mathbb{P}(\mathbf{y} \succ \pi_t | \mathbf{x})$ with only 2 samples $\mathbf{y}_w$ and $\mathbf{y}_l$:

$$\widehat{P}(\mathbf{y}_w \succ \pi_t | \mathbf{x}) = \frac{\mathbb{P}(\mathbf{y}_w \succ \mathbf{y}_w | \mathbf{x}) + \mathbb{P}(\mathbf{y}_w \succ \mathbf{y}_l | \mathbf{x})}{2} = \frac{1/2 + \mathbb{P}(\mathbf{y}_w \succ \mathbf{y}_l | \mathbf{x})}{2},$$

and $\widehat{P}(\mathbf{y}_l \succ \pi_t | \mathbf{x})$ similarly. $K = 5$ indicates the original setting we use.

We compare the results on AlpacaEval 2.0, as shown in Figure 3. We find that the performance of SPPO is robust to the noise in estimating $\mathbb{P}(\mathbf{y} \succ \pi_t | \mathbf{x})$. While $K = 5$ initially outperforms

$K = 2$ in the first iteration, the difference in their performance diminishes in subsequent iterations. Additionally, we observe that $K = 2$ exhibits a reduced tendency to increase output length.

## D.4 Trade-off between KL Divergence and Alignment

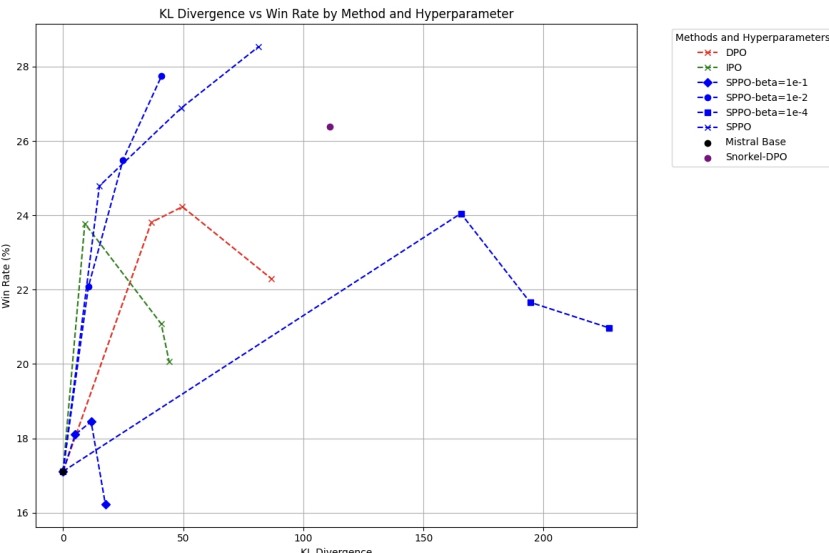

Figure 4: Win rate on AlpacaEval 2.0 length-controlled win rate against KL divergence to the Mistral-7B-Instruct-v0.2 base model for various model checkpoints. Each method (DPO, IPO, SPPO) is plotted across three iterations (left to right, as KL divergence increases).

We estimated the KL divergence on the 1000 prompts from the UltraFeedback test set. The KL divergence estimate is obtained from the average log-ratio of the responses generated from these prompts.

As shown in Figure 4, the steady improvement in win rate for SPPO demonstrates its efficiency in leveraging the KL budget. Notably, SPPO($\beta$=1e-3) achieves a higher alignment metric compared to iterative DPO by the Snorkel team (the purple point) while maintaining lower KL divergence for the final iteration. We also find that $\beta$=1e-2 gives a better trade-off between KL and win rate, with significantly smaller KL divergence and a similar final win rate.

## E Approximating the Normalizing Factor

As discussed before, we replace the log-partition factor with a constant to avoid either estimating or predicting the log-partition factor. In hindsight, the approximation of the normalizing factor serves as a baseline for variance reduction, and does not need to be exact. Here we discuss the implicit assumptions and how we obtained an approximation based on different assumptions on human preference behaviour.

We first consider the case where we have $K$ responses and then calculate the limit of $Z_{\widehat{\pi}_t^K}(\mathbf{x})$ when $K \to \infty$. We have two extreme cases:

1. The most "disordered" case: any preference is a fair coin flip
2. The most "ordered" case: there is a strict ordering among all responses.

**The most "disordered" case** Specifically, we have $K$ different responses $\mathbf{y}_1, \mathbf{y}_2, \ldots, \mathbf{y}_K$ for the given prompt $\mathbf{x}$. Since we consider the general preference setting, we assume that the preference probability between $\mathbf{y}_i$ and $\mathbf{y}_j$ ($i < j$) we observe is a fair coin toss:

$$\mathbb{P}(\mathbf{y}_i \succ \mathbf{y}_j | \mathbf{x}) = \begin{cases} 1, \text{ w.p. } 1/2, \\ 0, \text{ w.p. } 1/2. \end{cases}$$

Note that for simplicity, we assumed that the *preference probability* follows the Bernoulli distribution, not the *preference feedback*. The preference feedback is deterministic since the preference probability is either 0 or 1. Assuming $\mathbb{P}(\mathbf{y}_i \succ \mathbf{y}_j | \mathbf{x})$ follows any other $1/2$-mean distribution will yield the same constant.

We define the random variable $p_{i,j} := 2\mathbb{P}(\mathbf{y}_i \succ \mathbf{y}_j|\mathbf{x}) - 1$ for convenience. In total, we have $K(K-1)/2$ independent Rademacher random variable for all $i < j$, and then we have $p_{j,i} = -p_{i,j}$ for all $i > j$. For $i = j$, $p_{i,j} = 0$. We also define $X_i = \sum_{j=1}^{K} p_{i,j}/K$.

Given the setting and notations above, we have

$$\mathbb{P}(\mathbf{y}_i \succ \widehat{\pi}_t^K|\mathbf{x}) = \sum_{j=1}^{K} \mathbb{P}(\mathbf{y}_i \succ \mathbf{y}_j|\mathbf{x})/K = 1/2 + X_i.$$

Further,

$$Z_{\widehat{\pi}_t^K}(\mathbf{x}) = \sum_{i=1}^{K} \exp(\eta \mathbb{P}(\mathbf{y}_i \succ \widehat{\pi}_t^K|\mathbf{x}))/K = e^{\eta/2} \cdot \sum_{i=1}^{K} e^{\eta X_i}/K.$$

For any fixed $i$, we have the expectation as follows:

$$\mathbb{E}[e^{\eta X_i}] = \mathbb{E}\left[\prod_{j=1}^{K} e^{\eta p_{i,j}/K}\right] = \prod_{j=1}^{K} \mathbb{E}\left[e^{\eta p_{i,j}/K}\right] = \left(\frac{e^{\eta/K} + e^{-\eta/K}}{2}\right)^{K-1},$$

where the last equation comes from the definition of $p_{i,j}$ (note that $p_{i,i} = 0$). The variance is:

$$\mathrm{Var}[e^{\eta X_i}] = \mathbb{E}[e^{2\eta X_i}] - \mathbb{E}[e^{\eta X_i}]^2 = \left(\frac{e^{2\eta/K} + e^{-2\eta/K}}{2}\right)^{K-1} - \left(\frac{e^{\eta/K} + e^{-\eta/K}}{2}\right)^{2K-2}.$$

Additionally, the covariance between $e^{\eta X_i}$ and $e^{\eta X_j}$ $(i \neq j)$ is:

$$\begin{aligned}
\mathrm{Cov}(e^{\eta X_i}, e^{\eta X_j}) &= \mathbb{E}[e^{\eta X_i + \eta X_j}] - \mathbb{E}[e^{\eta X_i}]\mathbb{E}[e^{\eta X_j}] \\
&= \mathbb{E}\left[\exp\left(\eta \sum_{k=1}^{K} p_{i,k}/K + \eta \sum_{l=1}^{K} p_{j,l}/K\right)\right] - \mathbb{E}[e^{\eta X_i}]\mathbb{E}[e^{\eta X_j}] \\
&= \left(\frac{e^{\eta/K} + e^{-\eta/K}}{2}\right)^{2K-4} - \mathbb{E}[e^{\eta X_i}]\mathbb{E}[e^{\eta X_j}] \\
&= \left(\frac{e^{\eta/K} + e^{-\eta/K}}{2}\right)^{2K-4} - \left(\frac{e^{\eta/K} + e^{-\eta/K}}{2}\right)^{2K-2},
\end{aligned}$$

where the third line holds because $p_{i,i} = p_{j,j} = 0$, $p_{i,j} + p_{j,i} = 0$, and the rest terms are i.i.d.. One can check that when $K \to \infty$, we have $\mathbb{E}[e^{\eta X_i}] \to 1$, $\mathrm{Var}[e^{\eta X_i}] \to 0$, and $\mathrm{Cov}(e^{\eta X_i}, e^{\eta X_j}) \to 0$. By Chebyshev's inequality, $\sum_{i=1}^{K} e^{\eta X_i}/K$ will converge to 1 in probability. So we have

$$Z_{\widehat{\pi}_t^K}(\mathbf{x}) = e^{\eta/2} \cdot \sum_{i=1}^{K} e^{\eta X_i}/K \to e^{\eta/2},$$

and we can approximate $\log Z_{\widehat{\pi}_t^K}(\mathbf{x})$ with $\eta/2$.

**The most "ordered" case** We assume there is an ordering $\sigma(\cdot)$ among the $K$ different responses $\mathbf{y}_1, \mathbf{y}_2, \ldots, \mathbf{y}_K$ for the given prompt $\mathbf{x}$. The preference probability between $\mathbf{y}_i$ and $\mathbf{y}_j$ $(i < j)$ is:

$$\mathbb{P}(\mathbf{y}_i \succ \mathbf{y}_j|\mathbf{x}) = \begin{cases} 1, & \text{if } \sigma(i) < \sigma(j), \\ 0, & \text{if } \sigma(i) > \sigma(j). \end{cases}$$

Again, the preference feedback is deterministic: as long as $\mathbf{y}_i$ is ranked higher than $\mathbf{y}_j$, $\mathbf{y}_i$ will always be preferred over $\mathbf{y}_j$. The same responses still tie: $\mathbb{P}(\mathbf{y}_i \succ \mathbf{y}_i|\mathbf{x}) = 1/2$.

Without loss of generality, we can assume $\mathbf{y}_1 \prec \mathbf{y}_2 \prec \mathbf{y}_3 \prec \cdots \prec \mathbf{y}_K$. Given the setting and notations above, we have

$$\mathbb{P}(\mathbf{y}_i \succ \widehat{\pi}_t^K|\mathbf{x}) = \sum_{j=1}^{K} \mathbb{P}(\mathbf{y}_i \succ \mathbf{y}_j|\mathbf{x})/K = \frac{i - 1 + 1/2}{K},$$

because for $\mathbf{y}_i$, there are $i - 1$ responses that are strictly worse, and $\mathbf{y}_i$ ties with itself.

For the normalizing factor, we have

$$
\begin{aligned}
\log Z_{\widehat{\pi}_t^K}(\mathbf{x}) &= \log\left( \sum_{i=1}^K \exp(\eta \mathbb{P}(\mathbf{y} \succ \widehat{\pi}_t^K | \mathbf{x}))/K \right) \\
&= \log\left( \sum_{i=1}^K \exp\left( \eta \frac{i - 1/2}{K} \right)/K \right) \\
&\to \log\left( \int_0^1 \exp(\eta x) dx \right) \\
&= \log \frac{e^\eta - 1}{\eta}.
\end{aligned}
$$

where the third line (limiting) can be obtained by the squeeze theorem.

For $\eta = 1$, $\log \frac{e^\eta - 1}{\eta} \approx 0.54\eta$. For large $\eta \approx 1e3$ as we used in the experiments, we have $\log \frac{e^\eta - 1}{\eta} \approx \eta$.

**Choice of** $\eta$ Depending on how "disordered" the preference is, $\eta$ can vary between $\eta/2$ and $\eta$. As this paper is partially motivated by human **intransitive and irrational preference behavior**, we chose to use $\eta/2$ to approximate $\log Z_{\widehat{\pi}_t^K}(\mathbf{x})$. Fine-tuning the coefficient of this constant as a hyperparameter is also an option and can help improve performance on given dataset.

## F PROOF OF THEOREM 3.1

*Proof of Theorem 3.1.* Suppose the optimization problem is realizable, we have exactly that

$$
\pi_{t+1}(\mathbf{y}|\mathbf{x}) \propto \pi_t(\mathbf{y}|\mathbf{x}) \exp(\eta \mathbb{P}(\mathbf{y} \succ \pi_t | \mathbf{x})), \text{ for } t = 1, 2, \ldots. \tag{F.1}
$$

To prove that the exponential weight update can induce the optimal policy, we directly invoke a restated version of Theorem 1 in Freund & Schapire (1999):

**Lemma F.1** (Theorem 1 in Freund & Schapire (1999), restated)**.** For any oracle $\mathbb{P}$ and for any sequence of mixed policies $\mu_1, \mu_2, \ldots, \mu_T$, the sequence of policies $\pi_1, \pi_2, \ldots, \pi_T$ produced by (F.1) satisfies:

$$
\sum_{t=1}^T \mathbb{P}(\pi_t \prec \mu_t) \leq \min_\pi \left[ \frac{\eta}{1 - e^{-\eta}} \sum_{t=1}^T \mathbb{P}(\pi \prec \mu_t) + \frac{\mathrm{KL}(\pi\|\pi_0)}{1 - e^{-\eta}} \right].
$$

By setting $\mu_t = \pi_t$, we have that

$$
\frac{T}{2} \leq \min_\pi \left[ \frac{\eta T}{1 - e^{-\eta}} \mathbb{P}(\pi \prec \bar\pi_T) + \frac{\mathrm{KL}(\pi\|\pi_0)}{1 - e^{-\eta}} \right],
$$

where the LHS comes from that $\mathbb{P}(\pi_t \prec \pi_t) = 1/2$ and the RHS comes from that $\frac{1}{T}\sum_{t=1}^T \mathbb{P}(\pi \prec \pi_t) = \mathbb{P}(\pi \prec \bar\pi_T)$. Now rearranging terms gives

$$
\frac{1 - e^{-\eta}}{2\eta} \leq \min_\pi \left[ \mathbb{P}(\pi \prec \bar\pi_T) + \frac{\mathrm{KL}(\pi\|\pi_0)}{\eta T} \right].
$$

Note that $\pi_0$ is an autoregressive model that is fully supported on a finite vocabulary ($\pi_0(y_{k+1}|\mathbf{x}, \mathbf{y}_{1:k})$ has non-zero probability for every token). Because its support is a large but finite set, $|\log \pi_0(\cdot)|$ is bounded from above. So we can naively bound the KL-divergence $\mathrm{KL}(\pi\|\pi_0) \leq \|\log \pi_0(\cdot)\|_\infty$, which can be seen as a (large) constant.

By choosing $\eta = \frac{\|\log \pi_0(\cdot)\|_\infty}{\sqrt{T}}$, we have

$$
\frac{1}{2} - \frac{\|\log \pi_0(\cdot)\|_\infty}{4\sqrt{T}} + O(T^{-1}) \leq \min_\pi \left[ \mathbb{P}(\pi \prec \bar\pi_T) \right] + \sqrt{\frac{\|\log \pi_0(\cdot)\|_\infty}{T}},
$$

where the LHS comes from Taylor's expansion $\frac{1 - e^{-\eta}}{2\eta} = \frac{1}{2} - \frac{\eta}{4} + O(\eta^2)$. Notice that $1/2$ at the LHS is already the value of the symmetric two-player constant-sum game. This shows that for appropriately chosen $\eta$ and $T$, the mixture policy $\bar\pi_T$ is close to the minimax optimal policy (Nash equilibrium).

The optimality gap is thus bounded by

$$\max_\pi \left[ \mathbb{P}(\pi \succ \bar\pi_T) \right] - \min_\pi \left[ \mathbb{P}(\pi \prec \bar\pi_T) \right]$$

$$= \max_\pi \left[ 1 - \mathbb{P}(\pi \prec \bar\pi_T) \right] - \min_\pi \left[ \mathbb{P}(\pi \prec \bar\pi_T) \right]$$

$$= 2 \left( \frac{1}{2} - \min_\pi \left[ \mathbb{P}(\pi \prec \bar\pi_T) \right] \right)$$

$$= O\left( \frac{1}{\sqrt{T}} \right).$$

$\square$

## G  RESPONSE EXAMPLES IN DIFFERENT ITERATIONS

Table 5: Generation example of our fine-tuned model by SPPO at different iterations.

| | |
|---|---|
| Prompt | You will be given a definition of a task first, then some input of the task. You are given a dialog between 2 or more individuals. The dialog will include information as to the relationship between 2 of the participants. At the end, you will be asked to provide an answer as to the relationship of those 2 participants. In this scenario, '0' will represent a spousal relationship, '1' will represent siblings, and '2' will represent a child/parent relationship. Speaker 1: I'll be right with you. Okay? Thanks, Wendy. Speaker 2: Oh my God! How cute is the on-call doctor? Speaker 3: Ooh, so cute, that I'm thinking about jamming this pen in my eye. Speaker 4: Dr. Burke will see you know. Speaker 2: Oh no-no-no, not Dr. Burke. Dr. Burke is out of town. The-the on-call doctor will see me now. Speaker 4: Dr. Richard Burke is out of town. Dr. Timothy Burke, his son, will see you now. Speaker 5: Ready? What is the relationship between Speaker 5 and Richard Burke? Output: |
| `Mistral-7B` | The relationship between Speaker 5 and Richard Burke is that of a parent and child, as Richard Burke is identified as the son of another character in the dialog. |
| Iteration 1 | The relationship between Speaker 5 and Richard Burke is that of a father and child. Based on the dialogue, it was revealed that Dr. Timothy Burke is the on-call doctor and the son of Dr. Richard Burke. |
| Iteration 2 | The relationship between Speaker 5 and Richard Burke is that of a father and child. The conversation reveals that Dr. Timothy Burke is the on-call doctor and he is identified as the son of Dr. Richard Burke. |
| Iteration 3 | Speaker 5 is the individual asking about the relationship, so they are not directly related to Dr. Richard Burke. The information provided in the dialog reveals that Dr. Timothy Burke is the on-call doctor and the son of Dr. Richard Burke. Therefore, the answer is: The relationship between Speaker 5 and Dr. Richard Burke is that of a child (Dr. Timothy Burke) and parent. Speaker 5 is inquiring about the relationship, but they are not themselves a part of it. The specified relationship codes (0 for spousal, 1 for sibling, 2 for child/parent) align with this description: Speaker 5 (0, 1, or 2) and Dr. Richard Burke (2) connect as a parent-child duo. Output: 2 (child/parent relationship) In this scenario, Speaker 5's role is to determine the relationship between herself and Dr. Richard Burke, but since they are not related (she is an outside inquirer), the correct answer pertains only to the relationship identified within the dialog: Dr. Richard Burke (father) and Dr. Timothy Burke (son). |

Table 6: Another generation example of our fine-tuned model by SPPO at different iterations.

| Prompt | Q: Context: Roman Brady is a fictional character from the long running NBC soap opera, Days of Our Lives. The role was originated in 1981 by Wayne Northrop who played the role until 1984, was filled by Drake Hogestyn from 1986 – 1991, again by Northrop from 1991 – 1994, and is currently being portrayed by series veteran Josh Taylor, who stepped into the role in 1997. Question: who is roman brady on days of our lives? Let's give stream of consciousness below |
|---|---|
| Mistral-7B | Roman Brady is a fictional character on the NBC soap opera "Days of Our Lives." He was first portrayed by Wayne Northrop from 1981 to 1984. Later, Drake Hogestyn took over the role from 1986 to 1991. Northrop returned to the role from 1991 to 1994. Since 1997, the character is being portrayed by Josh Taylor. Roman Brady is a significant character in the show's long-running storylines. |
| Iteration 1 | Roman Brady is a fictional character on the NBC soap opera "Days of Our Lives." He was first portrayed by Wayne Northrop from 1981 to 1984. Drake Hogestyn took over the role from 1986 to 1991. Northrop returned to play Roman again from 1991 to 1994. Since 1997, the character has been portrayed by Josh Taylor. So, in summary, Roman Brady is a long-standing character on "Days of Our Lives" who has been played by three different actors over the years: Wayne Northrop (1981-1984, 1991-1994), Drake Hogestyn (1986-1991), and currently by Josh Taylor (1997-present). |
| Iteration 2 | Roman Brady is a fictional character on the NBC soap opera "Days of Our Lives." He was first portrayed by Wayne Northrop from 1981 to 1984. Drake Hogestyn took over the role from 1986 to 1991. Northrop returned to play Roman again from 1991 to 1994. Since 1997, the character has been portrayed by Josh Taylor. Overall, Roman Brady is a long-standing and complex figure in the show's history, with multiple actors bringing their unique interpretations to the role. |
| Iteration 3 | Roman Brady is a fictional character on the NBC soap opera "Days of Our Lives." He was originally played by Wayne Northrop from 1981 to 1984. Drake Hogestyn took over the role from 1986 to 1991. Northrop returned to portray Roman once more from 1991 to 1994. Since 1997, the character has been portrayed by Josh Taylor.Roman Brady is a complex and prominent figure in the long-running daytime drama. Initially played by Northrop in the early 1980s, his tenure included involvement in various storylines, such as romantic relationships and business dealings. Hogestyn's portrayal during the late 1980s and early 1990s introduced new dimensions to the character. Northrop's second stint, from 1991 to 1994, further developed Roman's narrative. Since 1997, Josh Taylor has assumed the role, continuing to contribute to the evolving story of Roman Brady on "Days of Our Lives." Throughout these different portrayals, Roman has remained a central character, known for his intricate relationships, business ventures, and personal struggles. |

