# OpenReview forum: "Self-Play Preference Optimization for Language Model Alignment"
_ICLR.cc/2025/Conference — ICLR 2025 Poster_

### Official Review · Reviewer_fKEZ · 2024-10-31

**Soundness:** 3
**Presentation:** 3
**Contribution:** 3
**Rating:** 6
**Confidence:** 4

**Summary:**

The work proposes a new method of LLM alignment called Self-Play Preference Optimization (SPPO). This algorithm is an iterative process. In each iteration, the Mean Square Error (MSE) loss function is minimized between the implicit reward formulated by Rafailov [1], in which the policy from the previous iteration acts as the reference policy, and the reward normalized by a function Z (x), which depends on the prompt (in practical implementation, this function is replaced by a constant).

Based on works that consider online alignment methods from the perspective of game-theory,  the expected probability that the response y is better than the responses of the policy from the previous iteration is used as reward for the answer y (in a practical implementation, the expectation is estimated by a finite number of responses using a preference model).

Proof of convergence to the Nash equilibrium on average is demonstrated. The proposed algorithm's connection with the Policy Gradient (PG) algorithm and Token-level Q* learning is shown.

The experimental setup used Llama-3-8B-Instruct and Mistral-7B-Instruct-v0.2 models, UltraFeedback dataset, and PairRM was used as the preference model. The quality was evaluated using AlpacaEval 2.0, Arena-Hard, MT Bench, and OpenLLM LeaderBoard benchmarks. The results show that SPPO generally outperforms their baseline counterparts.

**Strengths:**

- Convergence: A proof of algorithm convergence is provided.
- Relation to Other Methods: Theoretical analysis demonstrates how the proposed algorithm relates to other popular methods.
- Benchmarks: The algorithm's performance is evaluated on a large number of popular benchmarks for assessing general assistants.

**Weaknesses:**

The main weaknesses of this paper are threefold:

- Missing Analysis of Hyperparameter $\eta$ ($\beta$ in Rafailov et al. [1]): Rafailov et al. [2] demonstrated a clear relationship between the KL divergence with the original policy and the quality of responses. Comparing algorithms with different KL divergences is incorrect, as one may have lower response quality due to a lesser KL divergence with the original policy. Quality can be improved by adjusting the hyperparameter  $\eta $ ($\beta$). The results would be more convincing if presented as a Pareto front, as done by Rame et al. [3].
- Absence of Comparison with Other Game-Theory-Based Methods: Since the work relies on Nash equilibrium and cites other methods grounded in the same theory, it is logical to compare the proposed method with the Nash-MD [4] algorithm, as well as Nash-EMA [4] and DNO [5] (For Nash-MD and Nash-EMA the convergence was proven by the last iteration rather than on average).
- Limited Experimental Setup: The algorithm's efficacy was tested only on two models of approximately the same size and one dataset. It would be interesting to see how the proposed method behaves with models of different sizes and on other tasks.

References :
[1] Rafailov et al, Direct Preference Optimization: Your Language Model is Secretly a Reward Model.
[2] Rafailov et al, Scaling Laws for Reward Model Overoptimization in Direct Alignment Algorithms.
[3] Rame et al, WARP: On the Benefits of Weight Averaged Rewarded Policies.
[4] Munos et al, Nash Learning from Human Feedback.
[5] Rosset et al, Direct Nash Optimization: Teaching Language Models to Self-Improve with General Preferences.

**Questions:**

Am I correct in understanding that the equality between equations 3.9 and 3.10 is not strict and is accurate up to a constant $\frac{2}{\eta}$?

---

> ### Author Response · Authors · 2024-11-23
>
> We would like to thank the reviewers for their thoughtful feedback and valuable suggestions. Below, we address each of the raised questions in detail.
>
> ---
>
> > **Q1**: Missing Analysis of Hyperparameter η (β) in [1]): [2] demonstrated a clear relationship between the KL divergence with the original policy and the quality of responses. Comparing algorithms with different KL divergences is incorrect, as one may have lower response quality due to a lesser KL divergence with the original policy. Quality can be improved by adjusting the hyperparameter η(β). The results would be more convincing if presented as a Pareto front, as done by Rame et al. [3].
>
> **A1**: Thanks for your suggestion, we will add discussion on [3]. We have uploaded a new figure in the updated version (Section D.4). We plot the AE2.0 LC win rate vs. the KL divergence to Mistral-7B-Instruct-v0.2, for each model checkpoint we reported in the paper, as well as Mistral-SPPO with different learning rate(1e-1, 1e-2, 1e-4; 1e-3 reported in the original submission). Notably, SPPO has the following results (for more results please see Section D.4):
>
> | model                                          |   kl_divergence |   win_rate |
> |:-----------|----------:|---------:|
> | Mistral-7B-Instruct-v0.2 (ref model) |            0    |      17.11 |
> | Snorkel-Mistral-PairRM-DPO           |          111.09 |      26.39 |
> | SPPO-Mistral-Iter1 (beta=1e-3)                       |           15.14 |      24.79 |
> | SPPO-Mistral-Iter2 (beta=1e-3)                       |           49.25 |      26.89 |
> | SPPO-Mistral-Iter3 (beta=1e-3)                       |           81.25 |      **28.53** |
> | SPPO-Mistral-Iter1 (beta=1e-2)                       |           10.57 |      22.08 |
> | SPPO-Mistral-Iter2 (beta=1e-2)                       |           24.97 |      25.49 |
> | SPPO-Mistral-Iter3 (beta=1e-2)                       |           41.00 |      **27.75** |
>
> Our understanding is that the reviewer may suspect that the high alignment metric of SPPO comes from a larger KL budget. The plot (and table here) shows that this is not the case. Compared with other methods, the three iterations of SPPO do not incur more KL divergence but have shown a steady win rate increase. For the last iteration, it has a higher alignment metric, but less KL divergence compared with the iterative DPO by the Snorkel team. From these experiments, we also find that $\beta=1e-2$ gives a better trade-off between KL and win rate (almost half the KL but close win rate).
>
> ---
>
> > **Q2**: Absence of Comparison with Other Game-Theory-Based Methods: Since the work relies on Nash equilibrium and cites other methods grounded in the same theory, it is logical to compare the proposed method with the Nash-MD [4] algorithm, as well as Nash-EMA [4] and DNO [5] (For Nash-MD and Nash-EMA the convergence was proven by the last iteration rather than on average).
>
>
> **A2**: Thanks for your suggestion. We have added more discussion on the comparison with Nash-MD/Nash-EMA and DNO in the updated version (Section A). Here are some key points:
>
> - **DNO**: DNO [5] requires a gold response generated directly from GPT-4 and asks GPT-4 as a judge to assign a numerical score to each response for preference pair construction, which is a stronger signal than the preference feedback used in our work. This is quite a different task setting: access to a stronger language model’s generation motivates model distillation rather than model alignment, where only preference feedback is available. It would be hard to attribute whether the preference signal or the gold response helped alignment.
>
>   If we examine DNO without a gold response, then the practical version of DNO uses the DPO loss and the algorithm effectively becomes iterative DPO. It is therefore difficult to make a fair comparison between DNO and SPPO in experiments.
>
> - **Nash-MD/Nash-EMA**: We agree that these methods are among the first to consider alignment toward Nash equilibrium. The main difficulty in experimental comparison is that they may require calling $K^2$ PairRM comparisons to estimate the win rate as a reward on the fly. Unlike directly calling a reward model for each trajectory (prompt + response), the on-the-fly $K^2$ PairRM calls are computationally much heavier and may stall the training process due to their huge compute overhead. We are not aware of any open-sourced implementation that can perform real-time on-policy sampling and win rate calculation for 7B models.
>
>   We have emphasized the contribution of Nash-MD/EMA in the updated version and discussed their advantages/disadvantages. While Nash-MD enjoys last-iterate convergence, the theorem requires exact sampling from the mixture distribution $\pi_t(y|x)^{1-\alpha} \pi_{\text{ref}}^{\alpha}(y|x)$. This can only be done exactly in the bandit setting. For sequential generation, this mixture distribution is difficult to sample from and is approximated by token-level distribution mixing.

---

> > ### Author Response · Authors · 2024-11-23
> >
> > (continued)
> >
> > > **Q3**: Limited Experimental Setup: The algorithm's efficacy was tested only on two models of approximately the same size and one dataset. It would be interesting to see how the proposed method behaves with models of different sizes and on other tasks.
> >
> > **A3**: Thanks for your suggestion. We agree that it is worth examining SPPO with models of different sizes and on different datasets.
> >
> > **Policy models of different sizes** Due to computation resource limits, we were not able to fine-tune models as large as 70B. Here we present a larger model Gemma-2-9B:
> >
> > | Model                      | AlpacaEval 2.0 LC Win Rate | AlpacaEval 2.0 Win Rate |
> > |----------------------------|----------|----------|
> > | Gemma-2-9B-It              | 45.08    | 35.62    |
> > | Gemma-2-9B-SPPO Iter1      | 48.70    | 40.76    |
> > | Gemma-2-9B-SPPO Iter2      | 50.93    | 44.64    |
> > | Gemma-2-9B-SPPO Iter3      | 53.27    | 47.74    |
> >
> > SPPO shows consistent improvement, and can achieve higher win rate than GPT4, when the base model is larger and stronger.
> >
> > **Preference models of different sizes** We adopt several reward/preference models from [6], to Llama-3-8B-Instruct as the base model:
> >
> > - A strong Llama-3-8B BT reward model with 91.1% accuracy on Reward-Bench.
> > - A stronger Llama-3-8B general preference model with 92.2% accuracy on Reward-Bench (ranked 4th within 8B models, best 8B model is 93.1%).
> > - Smaller Gemma-2B-it-based reward/preference models.
> >
> > | Size  | Type | Iter | LC. WR | WR     | Avg. Len |
> > |-------|------|------|--------|--------|----------|
> > | base  |      |      | 23.07  | 23.34  | 1959     |
> > | 2B    | BT RM| 1    | 33.71  | 30.88  | 1843     |
> > |       |      | 2    | 37.38  | 37.88  | 2028     |
> > |       |      | 3    | 38.86  | 42.98  | 2244     |
> > |       | GPM  | 1    | 32.64  | 35.12  | 2102     |
> > |       |      | 2    | 35.16  | 41.40  | 2318     |
> > |       |      | 3    | 35.30  | 45.44  | 2490     |
> > | 8B    | BT RM| 1    | 33.38  | 29.90  | 1802     |
> > |       |      | 2    | 38.83  | 37.82  | 1945     |
> > |       |      | 3    | 40.55  | 37.09  | 1832     |
> > |       | GPM  | 1    | 33.39  | 31.49  | 1878     |
> > |       |      | 2    | 37.35  | 37.60  | 2020     |
> > |       |      | 3    | 39.72  | 39.38  | 1994     |
> >
> >
> >
> > With these different reward/preference models, SPPO has shown steady improvement and high performance on various evaluation benchmarks. Under other unchanged conditions, Llama-3-8B-Instruct + 8B BT reward model can achieve a 40.55% length-controlled win rate on AlpacaEval 2.0, while Llama-3-8B-Instruct + PairRM has 38.77%.
> >
> > **A different prompt dataset** Additionally, without any experimental setting changed, we run SPPO with a different prompt set berkeley-nest/Nectar, which is also used in a few previous works on LLM alignment (Starling-RM series, REBEL). The results are shown below.
> >
> > | Model                                      | Length Controlled Winrate | Win Rate | Standard Error | N Total | Avg Length |
> > |-------------------------------------------|---------------------------|----------|----------------|---------|------------|
> > | Mistral-7B-Instruct-SPPO-Nectar-Iter1     | 21.81                     | 20.63    | 1.20           | 805     | 1836       |
> > | Mistral-7B-Instruct-SPPO-Nectar-Iter2     | 22.29                     | 23.30    | 1.25           | 805     | 2092       |
> > | Mistral-7B-Instruct-SPPO-Nectar-Iter3     | 25.33                     | 27.12    | 1.32           | 805     | 2321       |
> >
> > SPPO still shows steady improvement on the AlpacaEval2.0 benchmark. Note that the prompt distribution is different from Ultrafeedback and we do not change any hyperparameters, so it is expected to see a slight drop in the LC win rate.
> >
> > ---
> >
> > > **Q4**: Am I correct in understanding that the equality between equations 3.9 and 3.10 is not strict and is accurate up to a constant $2\eta$?
> >
> > **A4**: Yes, there should be a $2\eta$ due to differentiation. We have corrected it in the updated version.
> >
> > ---
> >
> > **References**:
> > [1] Rafailov et al, Direct Preference Optimization: Your Language Model is Secretly a Reward Model.
> > [2] Rafailov et al, Scaling Laws for Reward Model Overoptimization in Direct Alignment Algorithms.
> > [3] Rame et al, WARP: On the Benefits of Weight Averaged Rewarded Policies.
> > [4] Munos et al, Nash Learning from Human Feedback.
> > [5] Rosset et al, Direct Nash Optimization: Teaching Language Models to Self-Improve with General Preferences.
> > [6] General Preference Modeling with Preference Representations for Aligning Language Models (https://arxiv.org/abs/2410.02197)
> >
> > ---

---

> > > ### Author Response · Authors · 2024-11-25
> > > **We're looking forward to your reply!**
> > >
> > > Dear Reviewer fKEZ,
> > >
> > > We appreciate your constructive review of our work and have provided detailed responses to your comments. As the discussion period is ending soon, we wanted to check if there are any remaining concerns we can address.
> > >
> > > We hope that with these clarifications and updates, you might consider revisiting your assessment of our submission.
> > >
> > > We look forward to your reply and any further discussion.
> > >
> > > Best regards,
> > > Authors

---

> > > > ### Comment · Reviewer_fKEZ · 2024-11-26
> > > > **Response**
> > > >
> > > > I've updated my score. Thank you for your feedback and additional experiments. Good luck!

---

> > > > > ### Author Response · Authors · 2024-11-26
> > > > > **Thank you!**
> > > > >
> > > > > Thank you for reassessing our work and for your valuable feedback, which helped us improve the paper. We truly appreciate your thoughtful review and support.

---

### Official Review · Reviewer_o5ze · 2024-11-04

**Soundness:** 3
**Presentation:** 3
**Contribution:** 3
**Rating:** 6
**Confidence:** 3

**Summary:**

This paper proposes a new preference learning algorithm, self-play preference optimization (SPPO for short), the main contribution comes from the new loss objective, which does not rely on pairwise comparisons. Experimental results show consistent improvements over iterative DPO and IPO.

**Strengths:**

1.	SPPO casts the RLHF as a constant-sum two-player game, and solves Nash equilibrium policy by proposing a simple form of loss function.
2.	This paper also draws some connections with policy gradient and token-level Q* learning approaches.

**Weaknesses:**

1.	The annotation step in line 4 of Algorithm 1 requires K^2 calls for preference model, which should be time consuming.
2.	The preference model in this paper should play a critical role, however, this paper only test on a very specific model, called PairRM, it is not clear to me how reliable this algorithm works on other tasks. Also, SPPO should be an online algorithm, the preference model should be also updated together with policy model in order to capture the probability distribution of evolving policy.
3.	Experiments are on prompts from Ultrafeedback, LLMs of 7-8B, and only one fixed preference model PairRM, SPPO should be tested in more general settings.

**Questions:**

1.	I believe the preference model, here is PairRM, is critical for the final performance of SPPO, it will be good if the authors also run experiments with other models (e.g. UltraRM-13B)
2.	The annotation step in line 4 of Algorithm 1 requires K^2 calls for preference model, then, the authors select the highest and lowest PairRM scores and winning and losing responses. This setting is used also for DPO and IPO? If so, the only differences come from the loss function definition?
3.	The work of SPO from Swamy et al. uses win rate as reward, and SPPO also computes K^2 preference scores, which should be very easy to be converted into win rate, then you can apply PPO etc for update. I’d like to see those results in Table 1 (this comparison can also answer some comments and questions from paragraph 4 in Section 1.)
4.	Snorkel-Mistral-PairRM-DPO should be a fair comparison for SPPO, as they both use PairRM in their pipeline. What’s the difference between Snorkel-Mistral-PairRM-DPO and Mistral-7B-DPO? The gap is very significant.
5.	In Figure 1 (should be Table?), why the base policy Mistral-7B-Instruct-v0.2 becomes worse at the first and second iterations. And the difference between 7.59 and 7.51 should be not significant.
6.	Do authors also evaluate Snorkel-Mistral-PairRM-DPO in Table 3? And Mistral-7B-DPO and IPO are worse than their base policy model. Thus, the prompts from Ultrafeedback should be not useful for GSM8k and other tasks?

---

> ### Author Response · Authors · 2024-11-23
>
> We would like to thank the reviewers for their detailed feedback and valuable suggestions. Below, we address each of the raised questions.
>
> ---
>
> > **Q1**: The annotation step in line 4 of Algorithm 1 requires K^2 calls for preference model, which should be time-consuming.
>
>
> **A1**: We agree that $O(K^2)$ calls are required. But calculating the preference score actually only requires one-step inference (rather than sequential generation) and can be performed in a batched and parallel way. So the time consumption can be alleviated given more computing resources. On the other hand, there are new techniques developed recently so that calculating $K^2$ pairwise preferences only requires $O(K)$ compute and can be as fast as a BT reward model (see [2]).
>
> We want to reiterate that if we assume there are nuances in human preference relations (such as non-transitivity), it is statistically inevitable to query the full preference matrix and pay $O(K^2)$ comparisons (see [1]).
>
> ---
>
> > **Q2**: The preference model should be also updated together with policy model in order to capture the probability distribution of evolving policy.
>
>
> **A2**: We would like to thank the reviewer for their insightful suggestion that the preference model can evolve jointly with the LLM policy. We believe this is an important further direction. In this work, the preference model is assumed to be the ground truth we aim to align with. Ideally, the pairwise preference should come from the human annotators, which is considered static for any given pair, so it does not have to evolve along with the LLM policy.
>
> In practice, we have to compromise, replacing humans with LLM-based annotators such as GPT-4, or PairRM in this case. We used PairRM mainly because it was state-of-the-art when the experiment was initially conducted, and our baseline Snorkel used PairRM and DPO with the same training data. For a fair comparison, we use PairRM as the ground truth.
>
> ---
>
> > **Q3**: I believe the preference model, here is PairRM, is critical for the final performance of SPPO, it will be good if the authors also run experiments with other models.
>
> **A3**: Thanks for your suggestion. We agree that it is worth examining SPPO with stronger reward models or preference models. We run SPPO with a stronger BT reward model and a general preference model from [2]:
>
> - A strong Llama-3-8B BT reward model with 91.1% accuracy on Reward-Bench.
> - A stronger Llama-3-8B general preference model with 92.2% accuracy on Reward-Bench (ranked 4th within 8B models, best 8B model is 93.1%).
>
> The results are:
> | Size  | Type | Iter | LC. WR | WR     | Avg. Len |
> |-------|------|------|--------|--------|----------|
> | base: Llama-3-8B-Instruct  |      |      | 23.07  | 23.34  | 1959     |
> | 8B    | BT RM| 1    | 33.38  | 29.90  | 1802     |
> |       |      | 2    | 38.83  | 37.82  | 1945     |
> |       |      | 3    | 40.55  | 37.09  | 1832     |
> |       | GPM  | 1    | 33.39  | 31.49  | 1878     |
> |       |      | 2    | 37.35  | 37.60  | 2020     |
> |       |      | 3    | 39.72  | 39.38  | 1994     |
>
>
> With these different reward/preference models, SPPO has shown steady improvement and high performance on benchmarks. Other conditions unchanged, Llama-3-8B-Instruct + 8B BT reward model can achieve a 40.55% length-controlled win rate on AlpacaEval 2.0, while Llama-3-8B-Instruct + PairRM has 38.77%.
>
> ---
>
> > **Q4**: The annotation step in line 4 of Algorithm 1 requires K^2 calls for preference model, then, the authors select the highest and lowest PairRM scores and winning and losing responses. This setting is used also for DPO and IPO? If so, the only differences come from the loss function definition?
>
>
> **A4**: That is right. Selecting the winner and loser to form a pair is adopted by the baseline method Snorkel. This setting is also used in our own implementation of SPPO, iterative DPO and IPO for a fair comparison. The only difference is the loss function definition, and for SPPO we utilize the numerical win rate.

---

> > ### Author Response · Authors · 2024-11-23
> >
> > (continued)
> >
> >
> > > **Q5**: The work of SPO from Swamy et al. uses win rate as reward, and SPPO also computes K^2 preference scores, which should be very easy to be converted into win rate, then you can apply PPO etc for update. I’d like to see those results in Table 1 (this comparison can also answer some comments and questions from paragraph 4 in Section 1.)
> >
> >
> > **A5**: Thanks for your suggestion. We would like to explain the difficulty in directly applying PPO with a general preference model in our setting (or SPO from Swamy et al.):
> >
> > 1. **In-training on-policy sampling**: The standard way of online PPO is to first sample one (or a batch of) prompts; then for each prompt $x$, generate $K$ responses; then call $K^2$ PairRM comparisons to estimate the win rate as reward; then follow PPO to calculate clipped policy gradient and gradient for the value function. Unlike directly calling a reward model for each trajectory (prompt + response), the $K^2$ PairRM calls are computationally much heavier and may stall the training process due to their huge compute overhead. We are not aware of any open-sourced implementation that can perform real-time on-policy sampling and win rate calculation.
> >
> > 2. **Actor-critic considerations**: To run these actor-critic methods, an additional value function model needs to be maintained, and the hyperparameter tuning of policy gradient methods is expected to be harder due to the widely perceived training instability and the interaction between the value model and policy model.
> >
> > 3. **Semi-online PPO**: A different choice is semi-online PPO, similar to what SPPO implements: generate all responses and calculate the win rate at the start of each round. Then PPO is applied to this new dataset in an offline fashion: trajectories and rewards are constructed ahead of time and the model parameter just follows the PPO gradient. We are not aware of any successful implementation of such offline PPO in the scenario of LLM alignment, not to mention open-sourced ones.
> >
> > As explained above, an iterative offline PPO or a fully online PPO with a general preference model has not been seen before and we do not expect to successfully implement it within a few days since the rebuttal begins.
> >
> > ---
> >
> > > **Q6**: In Figure 1, why the base policy Mistral-7B-Instruct-v0.2 becomes worse at the first and second iterations. And the difference between 7.59 and 7.51 should be not significant.
> >
> >
> > **A6**: We are not fully certain about the reason. MT-bench is a benchmark for testing multi-turn conversation and instruction-following ability. Our fine-tuning setting only focuses on single-round conversation and instruction following. Additionally, MT-bench only contains 80 multi-round questions. Therefore, it is not expected to see a significant correlation between MT-bench and other benchmarks.
> >
> > ---
> >
> > > **Q7**: Do authors also evaluate Snorkel-Mistral-PairRM-DPO in Table 3? And Mistral-7B-DPO and IPO are worse than their base policy model. Thus, the prompts from Ultrafeedback should be not useful for GSM8k and other tasks?
> >
> >
> > **A7**: We report Snorkel-Mistral-PairRM-DPO in Table 3 (Row 2). We agree that Ultrafeedback is not highly correlated with math tasks such as GSM8k, and improvement is not expected for every subtask.
> >
> > ---
> >
> >
> > [1] Borda Regret Minimization for Generalized Linear Dueling Bandits. ICML 2024
> > [2] General Preference Modeling with Preference Representations for Aligning Language Models (https://arxiv.org/abs/2410.02197)

---

> > ### Comment · Reviewer_o5ze · 2024-12-02
> > **Response**
> >
> > Thanks for providing the detailed explanation. I increase my contribution score to 3. Thanks!

---

### Official Review · Reviewer_5o2e · 2024-11-04

**Soundness:** 4
**Presentation:** 3
**Contribution:** 3
**Rating:** 6
**Confidence:** 4

**Summary:**

This paper introduces SPPO an approach to align language models with human preferences. SPPO models alignment as a two-player, constant-sum game aimed at reaching the Nash equilibrium, which provides a theoretically sound basis for iteratively optimizing preferences. The approach uses a smaller, pre-trained preference model PairRM and a limited dataset to achieve high alignment performance across various benchmarks, including AlpacaEval 2.0, MT-Bench, and the Open LLM Leaderboard, without additional supervisions from stronger models.

**Strengths:**

- Introduced a self-play, game-theoretical framework for preference alignment, leveraging a Nash equilibrium formulation.
- Proposed a new optimization objective that improves practical application and is theoretically motivated.
- SPPO demonstrates state-of-the-art results in multiple language model benchmarks and alignment benchmarks, outperforming DPO and IPO.

**Weaknesses:**

- Lack of comparison with conventional RLHF methods. This paper does not include a direct comparison with conventional RLHF methods, i.e., BT formulated RM and PPO training with it.
- Lack of comparison with similar methods. The paper lacks a comparison with conceptually similar work, such as Nash Preference Optimization (NPO). Without this, it is difficult to assess whether the proposed objectives in SPPO offer practical or theoretical advantages over existing approaches.

**Questions:**

The results indicate that the performance of DPO and IPO declines in iterations 2 and 3, whereas SPPO continues to improve across iterations. Could the authors provide insights why can SPPO maintains steady improvements while DPO and IPO do not?

---

> ### Author Response · Authors · 2024-11-23
>
> We would like to thank the reviewers for their thoughtful feedback and valuable suggestions. Below, we address each of the raised questions in detail.
>
> ---
>
> > **Q1**: The paper does not include a direct comparison with conventional RLHF methods, such as using a BT-formulated reward model (RM) and PPO training.
>
> **A1**: Thanks for the suggestion. We have conducted the following PPO+BT RM baseline:
>
> Policy: Mistral-7B-Instruct-v0.2
> Value function: Mistral-7B-Instruct-v0.2
> Reward Model: weqweasdas/RM-Mistral-7B (80.4% accuracy on Reward-Bench)
> Prompt Dataset: Ultrafeedback prompt.
>
> We use the standard implementation from the Huggingface trl library. The results are:
>
> | Model                               | LC Winrate | Win Rate | Std. | Avg Length |
> |-----|-----|-----|-----|-----|
> |Mistral-PPO                      |                       16.86  |   13.29        |    1.05   |   1371 |
>
> It is difficult to establish a fair comparison between SPPO (paired with a preference model) and PPO (paired with a reward model) because the quality of the preference or reward feedback significantly affects the performance of the learned LLM policy. For example, PPO here used a 7B Mistral-based reward model and SPPO used a 0.4B sequential classification model PairRM.
>
> We would also like to highlight the findings from [1], which demonstrate that certain mechanisms used in PPO—such as gradient clipping, value functions, and token reward redistribution—are unnecessary for LLM alignment. Instead, REINFORCE suffices. As discussed in Section 3.3, SPPO can be viewed as a multi-round, semi-online REINFORCE method. Within each round, SPPO essentially follows the policy gradient of REINFORCE. The key difference lies in that the reward (win rate) used by SPPO changes dynamically across rounds. We will add discussion on the connection with and difference from the conventional RLHF.
>
> [1] Admadian et al., *Back to Basics: Revisiting REINFORCE Style Optimization for Learning from Human Feedback in LLMs* (2024).
>
> ---
>
> > **Q2**: Lack of comparison with similar methods, such as Nash Preference Optimization (NPO). The paper lacks a comparison with conceptually similar work, such as Nash Preference Optimization (NPO). Without this comparison, it is difficult to assess the practical or theoretical advantages offered by SPPO.
>
> **A2**: We are not sure which specific paper is being referred to here. If you are referring to [2], our paper and DNO [2] are actually concurrent works, and we have discussed it in detail in Appendix A. The practical version of DNO employed the DPO loss, and the algorithm can be seen as iterative DPO. Furthermore, DNO requires a gold response directly generated by GPT-4, which provides a stronger signal compared to the preference feedback used in our work. DNO will eventually become iterative DPO in our setting for a fair comparison.
>
> [2] *Direct Nash Optimization: Teaching Language Models to Self-Improve with General Preferences*.
>
> ---

---

> > ### Comment · Reviewer_5o2e · 2024-11-29
> >
> > Thank you to the authors for providing the detailed explanation.
> >
> > I will maintain my positive assessment and adjust the soundness rating to better reflect the clarifications provided.

---

> > > ### Author Response · Authors · 2024-11-29
> > > **Thank you!**
> > >
> > > Thank you for raising the soundness rating and for providing valuable feedback that helped clarify the paper. We greatly appreciate your thoughtful review and support.

---

### Official Review · Reviewer_UKK9 · 2024-11-04

**Soundness:** 3
**Presentation:** 3
**Contribution:** 3
**Rating:** 6
**Confidence:** 4

**Summary:**

The paper proposes a novel self-play alignment method, SPPO, that leverages relative win rates instead of scalar-based reward models. SPPO aligns the language model without external pairwise preference pairs, highlighting its efficiency over pairwise alignment methods. Through iterative training with a theoretical guarantee of converging to the ideal policy, SPPO shows strong performance over previous methods.

**Strengths:**

1. SPPO is derived from a solid theoretical background with a theoretical guarantee on the convergence of training policy to the ideal policy.
2. Interpreting SPPO from several different perspectives (e.g., token-level Q-learning and policy gradient) further strengthens its validity.
3. SPPO led to monotonic improvements of both models in majority of tasks, highlighting the effectiveness of SPPO.

**Weaknesses:**

**1. Lack of comparisons against relative methods**

As the paper precisely depicts SPPO as a "semi-online" variant of the policy gradient method, SPPO has the potential to be interpreted as an online method by leveraging on-policy generations and having access to a preference model. Moreover, along with the foundational alignment methods like DPO and IPO, there are some strong offline methods like SimPO [1] that outperform IPO and DPO. However, any RL-based methods [2] or recent performant offline methods are compared to SPPO in the paper.

&nbsp;

**2. Insufficient analysis of empirical aspects of SPPO**

While SPPO's performance on diverse benchmarks is impressive, the paper does not suggest any analysis of how SPPO-trained models are aligned to PairPM. Especially regarding the choice of total training epochs of 18 and the iterative nature of SPPO, analysis of gradual shifts happening throughout the training would be needed, such as the loss trends or PairPM win rate changes.


&nbsp;

**3. Clarifications on terminologies for experiments**

Some experiment terminologies are not clearly stated. For instance, the concepts of "iteration", "round", or "epoch" are not clearly separated, so it is hard to follow the exact training procedure of SPPO in the paper. Related to this point, hyperparameters for DPO and IPO experiments throughout the iterations should also be more explicitly stated.


&nbsp;
&nbsp;


**References**

[1] SimPO: Simple Preference Optimization with a Reference-Free Reward (Meng et al., 2024)

[2] Back to Basics: Revisiting REINFORCE Style Optimization for Learning from Human Feedback in LLMs (Admadian et al., 2024)

**Questions:**

Along with some points above, some additional questions are:

**1. Generation length and SPPO?**

In common, DPO and SPPO are leveraging the sum of log ratio between $\pi$ and $\pi_t$ given the prompt and response pair $(x, y)$, which could be sensitive to the length of generation. As SPPO is a single response-based method, length variance in $\mathbf{y}\sim \pi_t(\cdot|\mathbf{x})$ could induce high variance in the SPPO loss, which could lead to spurious length biases during the optimization as in DPO [1]. Regarding SPPO also results in longer generations as in iterative DPO in Table 1, would SPPO also be prone to the length bias?

&nbsp;


**2. Compatibility with classifier RMs?**

As the paper thoroughly shows, PairRM best fits the algorithmic design of SPPO. However, the Bradley-Terry model-based reward models (i.e., classifier RMs) are also applicable to approximate $P(y > \pi_t|x)$ by comparing the scalar values. Furthermore, there are many classifier RMs which are probably better than PairRM on classifying the better/worse response regarding the moment of submission [2]. How would those RMs impact SPPO?


&nbsp;
&nbsp;


**References**

[1] Disentangling Length from Quality in Direct Preference Optimization (Park et al., 2024)

[2] RewardBench: Evaluating Reward Models for Language Modeling (Lambert et al., 2024)

---

> ### Author Response · Authors · 2024-11-23
>
> We would like to thank the reviewers for their thoughtful feedback and valuable suggestions. Below, we address each of the raised questions in detail.
>
> ---
>
> > **Q1**: The paper does not suggest any analysis of how SPPO-trained models are aligned to PairPM.
>
> **A1**: Thank you for your suggestion. In fact, we have reported the PairRM win rates from different iterations in Appendix D.3 (Figure 3). These results show that the PairRM win rate against the initial reference model (Mistral-7b-instruct-v0.2) consistently increases throughout the three iterations, demonstrating the effectiveness of SPPO.
>
> ---
>
> > **Q2**: The choice of total training epochs of 18.
>
> **A2**: We appreciate the question. We have updated the description in the paper for clarity. The 18 epochs were not fully utilized in actual training. We observed that stopping after the first iteration yielded the best performance, likely due to avoiding overfitting. The choice of 18 epochs was primarily for defining the linear-decay learning rate schedule.
>
> More specifically, the learning rate at step $t$ is computed as:
> $$
> \eta(t) = \bigg(1- \frac{t}{T} \bigg) \eta_0.
> $$
> Here $T=N*E$ is the planned total number of steps, where we have $N$ steps per epoch and $E$ epochs. We set $E=18$ but stop after the 1st epoch, and $18$ is only used when calculating the learning rate.
>
>
> ---
>
> > **Q3**: Terminology choices and hyperparameter statements.
>
> **A3**: Thank you for your feedback. We have revised the paper to improve clarity:
>
> - "Iteration" and "round" are used interchangeably and refer to the high-level updates indexed by $t = 1, 2, \dots, T$, meaning there are $T$ total rounds/iterations of approximating the exponential weight update rule. To avoid confusion, we have standardized the terminology to "iteration" throughout the paper.
> - "Epoch" describes the number of times each data sample is used within a single round of training. For example, 2 epochs indicate that each sample is utilized twice.
> - Regarding training hyperparameters, we provide the following details for DPO and IPO experiments:
>
>   - Iterative DPO:
>     - Rounds 1, 2, 3: \(\beta = 0.1\), learning rate = \(5e-7\), trained for 2 epochs per round.
>   - Iterative IPO:
>     - Rounds 1, 2, 3: \(\beta = 0.01\), learning rate = \(5e-7\), trained for 1 epochs per round.
>
> These details will also be updated in the appendix.
>
> ---
>
> > **Q4**: Would SPPO also be prone to the length bias?
>
> **A4**: We believe SPPO is less susceptible to length bias compared to other methods due to its loss function design, which constrains how far the log ratio can deviate. According to the proposed SPPO objective, $\log \frac{\pi_{\theta}(y_w|x)}{\pi_{\text{ref}}(y_w|x)}$ will at most increase to $\eta/2$, so the length will not increase arbitrarily.  In contrast, translation-invariant loss functions such as DPO and IPO cannot control the changes in the log ratio effectively but only the gaps between the winner and loser. Therefore, even when length bias exists in the dataset or preference oracle, SPPO’s loss formulation helps mitigate its impact.
>
> ---
>
> > **Q5**: How would better reward models/preference models impact SPPO’s performance?
>
> **A5**: SPPO is designed to work with any arbitrary preference model, including cyclic preference relations or Bradley-Terry (BT) models. Under the BT model, SPPO aims to maximize the win rate as defined by the BT model. Specifically, the win rate between two responses $y_i$ and $y_j$ can be computed as:
>
> $$
> P(y_i \succ y_j | x) = \text{sigmoid}(r(x, y_i) - r(x, y_j))
> $$
>
> Similarly, $P(y \succ \pi_t | x)$ can be estimated in the same way proposed in the paper. In theory, if the preference model follows the BT assumption, SPPO will converge to a policy that concentrates on responses with the highest BT reward.
>
> A stronger preference oracle—whether BT-based or not—will naturally enable SPPO to obtain stronger policies. For example, using a stronger Llama-3-8B BT reward model from [1] (91.1% accuracy on Reward-Bench) instead of PairRM led to better results:
>
> | Model | Iter | LC. WR | WR     | Avg. Len |
> |------|------|--------|--------|----------|
> | base: Llama-3-8B-Instruct   |      | 23.07  | 23.34  | 1959     |
> | SPPO w. BT RM| 1    | 33.38  | 29.90  | 1802     |
> |      | 2    | 38.83  | 37.82  | 1945     |
> |      | 3    | 40.55  | 37.09  | 1832     |
>
> Specifically, Llama-3-8B-Instruct with the 8B BT reward model achieved a 40.55% length-controlled win rate on AlpacaEval 2.0, outperforming the same model guided by PairRM (38.77%).
>
> ---
>
> **References**:
> [1] General Preference Modeling with Preference Representations for Aligning Language Models (https://arxiv.org/abs/2410.02197)

---

> > ### Comment · Reviewer_UKK9 · 2024-11-25
> >
> > I thank the authors for addressing the points mentioned in the review. While I still find the experiments quite limited, further details in the response helped me understand the paper better. Therefore, I will stay with my positive scores.

---

> > > ### Author Response · Authors · 2024-11-25
> > > **Thank you!**
> > >
> > > Thank you for taking the time to review our paper and for your positive feedback. We appreciate your acknowledgment of the additional details we provided in our response and are glad they helped clarify our work.
> > >
> > > Regarding your concern about the experiments, we would like to take this opportunity to provide further clarification:
> > >
> > > ---
> > > 1. **Analysis of SPPO alignment to PairPM:** To address how SPPO-trained models align with PairPM, we provide the PairRM win rate against the base model:
> > >
> > > | Model                           | PairRM Win Rate |
> > > |---------------------------------|-----------------|
> > > | Mistral-7B-Instruct-v0.2 (base model) | 50.0%          |
> > > | Mistral-7B-SPPO-Iter1           | 65.1%          |
> > > | Mistral-7B-SPPO-Iter2           | 72.2%          |
> > > | Mistral-7B-SPPO-Iter3           | 78.1%          |
> > >
> > >
> > > More detailed results and analyses can be found in Section D.2 of the paper.
> > >
> > > ---
> > > 2. **Comparison with stronger offline methods:** The key insight from SimPO is that averaged log-likelihood (implicit reward) can help mitigate length bias. This approach is orthogonal to what SPPO proposes. We are actively conducting new experiments to integrate the average log-likelihood into the SPPO objective and will update our findings as soon as the results are available.
> > >
> > >
> > > We hope this additional information addresses your concerns and further clarifies our approach. Thank you again for your thoughtful comments and support.

---

### Meta-Review · Area_Chair_WfB9 · 2024-12-20

**Metareview:**

The paper introduces SPPO, a self-play alignment method that leverages preference probabilities to overcome the limitations of traditional RLHF approaches. SPPO outperforming previous methods like DPO and IPO on benchmarks such as AlpacaEval 2.0, MT-Bench, and Arena-Hard. The paper's strengths include a solid theoretical foundation for SPPO with a convergence guarantee, multiple interpretations of SPPO to enhance its validity, effective demonstration of monotonic improvements in models across tasks.  However, the paper also has some weaknesses, such as the lack of comparisons with conventional online RLHF methods (e.g., PPO, RLOO), insufficient analysis of SPPO's empirical aspects, and unclear experimental terminologies. Some of these weaknesses were addressed during the rebuttal period, making this submission slightly above the bar for acceptance at ICLR.

Suggestion: While SPPO uses PairRM, a BT RM can also be used for comparing preference pairs, making it reasonable to compare SPPO with PPO, RLOO and other RLHF methods that use Bradley-Terry RMs.

**Additional Comments On Reviewer Discussion:**

The rebuttal period involved active discussion between the authors and reviewers, leading to significant improvements in the paper. The reviewers raised concerns about clarity, missing comparisons with other methods, potential length bias, the impact of different reward models, and the limited experimental setup.  The authors responded by providing clarifications on the experimental setup and additional experiments and results, comparing SPPO with conventional RLHF methods.  Furthermore, the authors provided evidence to argue that SPPO is less susceptible to length bias and demonstrated its compatibility with various preference models, including BT models.  Finally, they expanded their setup to include different models and datasets, demonstrating SPPO's effectiveness in more general settings.

---

### Decision · Program_Chairs · 2025-01-22

Accept (Poster)